# Evolutionary action of mutations reveals antimicrobial resistance genes in Escherichia coli

David C. Marciano [1,10 ✉], Chen Wang [1,10], Teng-Kuei Hsu[2], Thomas Bourquard [1], Benu Atri [3,9], Ralf B. Nehring [1,2,4,5], Nicholas S. Abel[6], Elizabeth A. Bowling[2], Taylor J. Chen[7], Pamela D. Lurie [1], Panagiotis Katsonis [1], Susan M. Rosenberg [1,2,4,5,7], Christophe Herman [1,4,5] & Olivier Lichtarge[1,3,5,8 ✉]

Since antibiotic development lags, we search for potential drug targets through directed evolution experiments. A challenge is that many resistance genes hide in a noisy mutational background as mutator clones emerge in the adaptive population. Here, to overcome this noise, we quantify the impact of mutations through evolutionary action (EA). After sequencing ciprofloxacin or colistin resistance strains grown under different mutational regimes, we find that an elevated sum of the evolutionary action of mutations in a gene identifies known resistance drivers. This EA integration approach also suggests new antibiotic resistance genes which are then shown to provide a fitness advantage in competition experiments. Moreover, EA integration analysis of clinical and environmental isolates of antibiotic resistant of *E. coli* identifies gene drivers of resistance where a standard approach fails. Together these results inform the genetic basis of de novo colistin resistance and support the robust discovery of phenotype-driving genes via the evolutionary action of genetic perturbations in fitness landscapes.

[1] Department of Molecular and Human Genetics, Baylor College of Medicine, Houston, TX 77030, USA. [2] The Verna and Marrs McLean Department of Biochemistry & Molecular Biology, Baylor College of Medicine, Houston, TX 77030, USA. [3] Structural and Computational Biology & Molecular Biophysics Program, Baylor College of Medicine, Houston, TX 77030, USA. [4] Department of Molecular Virology and Microbiology, Baylor College of Medicine, Houston, TX 77030, USA. [5] Dan L. Duncan Comprehensive Cancer Center, Baylor College of Medicine, Houston, TX 77030, USA. [6] Department of Pharmacology, Baylor College of Medicine, Houston, TX 77030, USA. [7] Integrative Molecular & Biomedical Biosciences Program, Baylor College of Medicine, Houston, TX 77030, USA. [8] Computational and Integrative Biomedical Research Center, Baylor College of Medicine, Houston, TX 77030, USA. [9] Present address: Clara Analytics Inc., 451 El Camino Real #201, Santa Clara, CA 95050, USA. [10] These authors contributed equally: David C. Marciano, Chen Wang. ✉email: david.marciano@bcm.edu; lichtarge@bcm.edu

The emergence of de novo antibiotic resistance is part of a slowly unfolding healthcare crisis wherein common infections, minor injuries and simple medical procedures could be deadly due to the infection of antibiotic-resistant strains. It is therefore important to find the basis of de novo antibiotic resistance to guide better treatment, to extend the life of current drugs, and to create drugs against new targets. For instance, although colistin is the drug of last resort against multi-drug resistant gram-negative bacterial infections[1], the emergence of a mobile colistin resistance gene (mcr) threatens its use[2]. Interestingly, despite the global spread of mcr genes, the mcr−1 gene made up only 4.6% of Parisian sequenced isolates that were resistant, suggesting other resistance mechanisms are present[3]. Hampering clinical sequencing from finding which mutations drive clinical outcomes are a combination of sequencing artifacts, poor coverage, imprecise reference genome and noisy mutational background[4,5]. The genetic basis of antibiotic resistance remains incomplete and slows progress towards better patient outcomes and new therapeutics for resistant bacteria[6].

In contrast to the uncontrolled variables inherent to natural populations, adaptive laboratory evolution (ALE) experiments provide precise control over the system (selection conditions, population size, genetics of the founding population) and the ability to store clones in stasis for later study. When coupled with the ability to sequence and edit DNA, ALE speeds the rigorous testing of specific mutations for their role in fitness[7–9]. Here, we develop a new method, called evolutionary action (EA) integration, to analyze the adaptive laboratory evolution of antibiotic-resistant E. coli in order to identify new genes contributing to colistin resistance.

Normally, when E. coli grows in an environment to which it is well-adapted, mutations are rare (a rate of ~$10^{-3}$/genome/generation)[10] and adaptive mutations are easily identified from parallel experiments since, in a large population, the probability of a same gene mutating in independent cultures is also small[11]. However, mutation rates are not actually uniform across the population and mutator clones that are DNA replication fidelity deficient often arise to greatly increase the number of mutations observed during an adaptation experiment[12–14]. This phenomenon is not idiosyncratic to laboratory selection as these mutators are also observed in clinical isolates and cancer cell populations undergoing natural selection[15,16]. In these highly mutagenic circumstances, fitness-improving mutations can become obscured by the large bulk of passenger mutations[12] which become genetically linked to beneficial mutations[17]. The inability to uncouple driver mutations from the background is in part due to the assumption that all mutations, prior to selection, have an equal probability of contributing to some phenotype. It is under these highly mutagenic situations that we apply our approach that differentially scores the mutational impact of mutations observed in a gene and determines the probability of seeing those mutations against a background of random mutations.

In order to differentially score nonsynonymous mutations we quantify the functional impact of coding mutations with evolutionary action (EA)[18]. The EA score of a mutation is a product of two terms. The first term accounts for the magnitude of an amino acid substitution at the position of interest. This is the step-size of the mutation in the fitness landscape, and it is approximated from substitution matrices (e.g. a serine to threonine substitution is more frequent and thus a smaller step in sequence space than a serine to tryptophan substitution). The second term is the functional sensitivity of a sequence position to amino acid substitutions. This sensitivity is the slope of the fitness landscape at that position, and it is approximated by Evolutionary Trace analysis, which ranks the positions of a protein sequence by the extent to which their variation correlate with large or with small

divergences in the phylogenetic tree of that protein family[19] (e.g. a position that varies only between distant evolutionary clades will have a large slope, and one that varies among evolutionary tree neighbor species will have a small slope). The product of magnitude times sensitivity, or step-size times slope, is the fitness effect of a mutation, which we call the evolutionary action (EA). Thus, EA interprets the functional impact of a coding variant in light of all past homolog variations and divergences tallied in sequence databases. Our hypothesis is that adaptive gene mutations necessarily impact function and thus will be biased to larger EA scores than random mutations.

In this work, we utilize the power of E. coli genetics to test this causal relationship between genes under selection pressure and elevated EA scores and to validate several new mediators of colistin resistance. These results lend support to the use of EA to unravel the genotype-phenotype relationship in fitness landscapes[20–22].

## Results

**ALE of ciprofloxacin and colistin resistant E. coli.** To thoroughly probe the fitness landscape, we passaged E. coli in the presence of either ciprofloxacin or colistin at three different mutation rates and sequenced the surviving populations (Fig. 1a). The base mutation rate of wild type E. coli MG1655 (WT) was elevated by either nucleotide analogs, 2-aminopurine and zebularine, (WT + mutagen)[23] or, genetically, through disabling mismatch repair (ΔmutL::zeoR) and mismatch proofreading from DNA polymerase III (mutD5; mutator)[24,25]. Within 18 days, most cultures displayed antibiotic resistance above the epidemiological cut-off of either ciprofloxacin (0.5 μg/mL) or colistin (2 μg/mL) (EUCAST) with some surviving at concentrations 1000-fold higher than initial inhibitory levels (Fig. 1b–g). DNA was extracted and sequenced from the evolved cultures and the nonsynonymous and nonsense mutations used in our analysis were determined (Supplementary Data 1, raw sequencing data available at SRA PRJNA543834).

We first estimated the coding region mutational rate to assess the mutational load of each condition. Note that we sequenced the evolved cultures, so more mutations are expected relative to sequencing individual clones from the cultures. The WT cultures average 0.2 coding region mutations per round of selection. Exposure to the mutagens raised the rate sevenfold to 1.4 mutations per round. In the mutator strains, the rate of mutation accumulation was extremely high with an average of 30.6 mutations per round for ciprofloxacin and 17.7 for colistin (Fig. 1h). The WT + mutagen and mutator regimes increased the mutational frequencies over wild type by approximately one and two orders of magnitude, respectively. At these higher rates, we anticipated most mutations would be hitch-hikers with minimal impact on fitness and thereby are expected to have low EA scores.

We next computed the evolutionary action (EA) scores corresponding to amino acid substitutions in E. coli K12 MG1655 protein-coding genes. For the sensitivity term of the EA, Evolutionary Trace (ET) scores of amino acid position importance were generated for 4169 of the 4374 proteins in E. coli MG1655 genome. Thereby, approximately 95% of genes have sequenced homologs available to yield ET scores. For the mutational magnitude term, the log-odds ratio of amino acid replacement was previously determined using 67,000 multiple sequence alignments and corresponding ET scores for proteins deposited in the Protein Data Bank[18].

Using the E. coli MG1655 EA scores, we established a random background distribution for mutations that have not undergone a selective pressure for enhanced fitness. We avoid direct selection pressure by collecting non-synonymous mutations from in-silico simulated random mutations. The EA distribution of

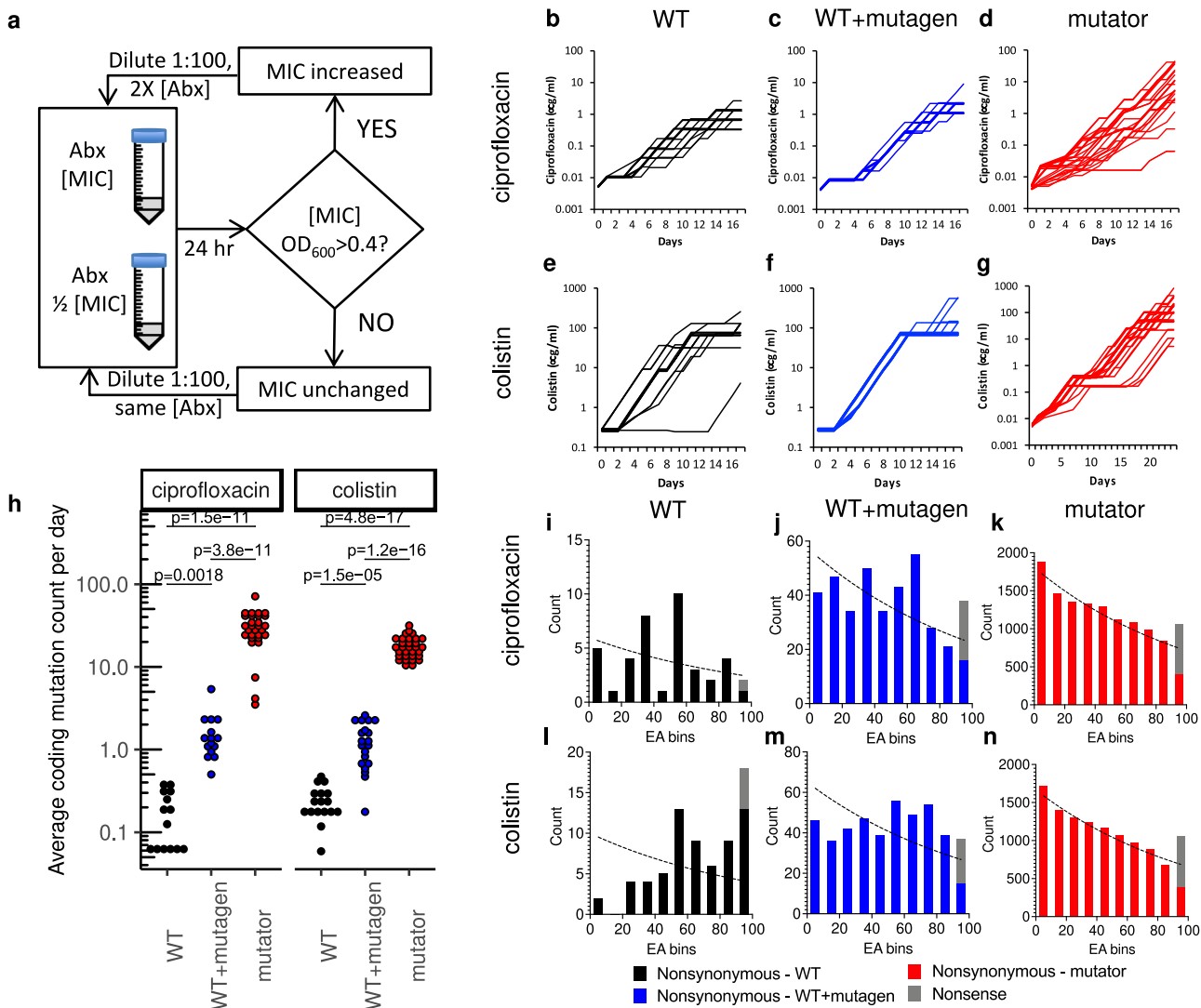

**Fig. 1 Experimental evolution. a** Flow chart showing how cultures are challenged to grow in minimum inhibitory concentrations (MIC) of an antibiotic (Abx) as determined by a culture optical density at 600 nm ($OD_{600}$) of greater than 0.4. **b–g** The change of ciprofloxacin dosage levels or colistin dosage for the adapting populations over time during the evolution experiment. The populations included (**b**) *E. coli* MG1655 (WT, black lines) grown in the presence of ciprofloxacin ($n = 14$), (**c**) *E. coli* MG1655 grown in the presence of nucleotide analogs (WT + mutagen, blue lines) and ciprofloxacin ($n = 15$), (**d**) a highly mutagenic *E. coli* mutD5 ΔmutL::zeoR (mutator, red lines) grown in the presence of ciprofloxacin ($n = 29$), **e** *E. coli* MG1655 grown in the presence of colistin ($n = 17$), (**f**) in the presence of nucleotide analogs and colistin ($n = 20$) and (**g**) *E. coli* mutD5 ΔmutL::zeoR grown in the presence of colistin ($n = 31$). Each culture's data set was iteratively offset 1% to help visualization of overlapping lines. **h** Each data point represents the number of nonsynonymous + nonsense mutations observed in whole-genome sequencing that has allele frequency above 0.1 in a sample divided by the day of collection (typically day 14). Statistical significance determined by two-tailed *t*-test with Bonferroni adjustment. Sample sizes and color scheme are the same as in panels **b–g**. **i–n** Distribution of EA scores observed across the populations sequenced in panels **b–g**. Bar colors correspond to panels **b–g** with the exception of nonsense mutations which are shown as stacked gray bars. The dashed curves represent the expected exponential decay curve generated from in silico random mutations (Supplementary Fig. 1a). Source data are provided as a Source Data file.

these non-synonymous mutations decays exponentially and is biased toward low impact EA scores (Supplementary Fig. 1a). This enrichment of low impact mutations is consistent with the inherent robustness in the genetic codon code, whereby biochemically similar amino acids are encoded by similar codons[26]. Comparing this background EA distribution to WT ALE samples under either ciprofloxacin or colistin selection shows an enrichment for high EA scores (Fig. 1i, l). Because these samples have few mutations, those which arise are expected to be impactful and play a role in adaptation to the antibiotic. In the WT + mutagen regime, the evolutionary action distributions shift towards lower EA scores which is consistent with an increase in random mutations (Fig. 1j, m). In the mutator strain samples, the distribution reverts to an

exponential decay pattern similar to the random background (Fig. 1k, n) suggesting that most of the observed mutations are hitchhikers and do not contribute to antibiotic resistance. The exponential decay pattern observed in the in silico and in the mutator strain is, as we discuss later, consistent with prior models of mutational displacements in the fitness landscape[27].

**EA integration predicts known antibiotic resistance genes.** The observation that randomly simulated nonsynonymous mutations trend towards low EA values while mutations observed in the ALE of MG1655(WT) tend to have high EA values supports our hypothesis that individual genes which contribute to antibiotic

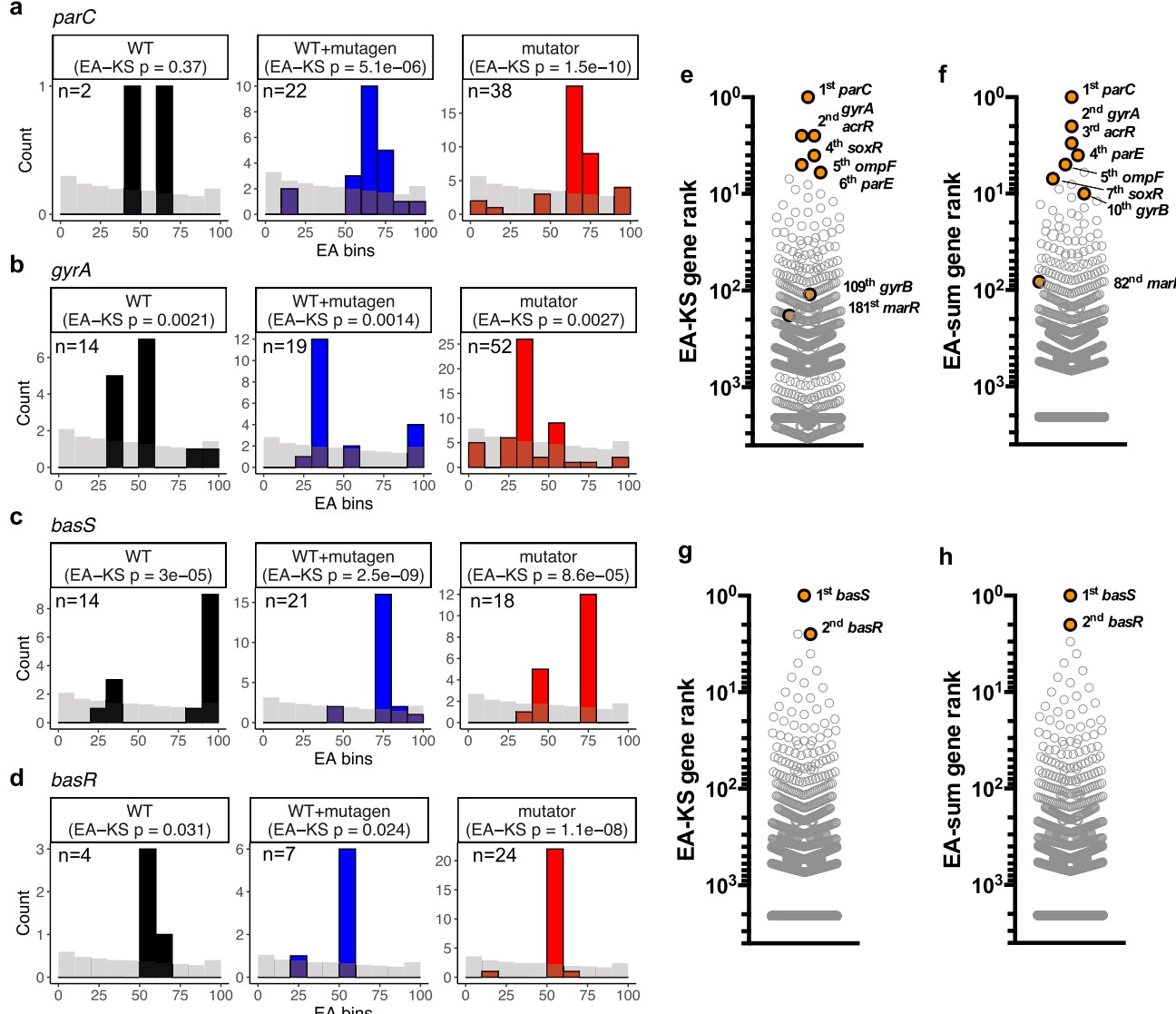

**Fig. 2 Known drivers of resistance have non-random EA integrals and are frequently mutated across cultures. a–d** The EA distributions of known drivers *parC*, *gyrA*, *basS* or *basR* mutations compared to randomly simulated mutations (gray bars, 63,507 randomly simulated coding mutations) with results of one-sided Kolmogorov-Smirnov (KS) test shown. The number of mutations observed in each gene and mutational conditions are label in the figure. For visual comparison, the simulated mutations bars were scaled to the mutation count observed for each gene in each mutation load. The *parC* and *gyrA* profiles are from the ciprofloxacin dataset while *basS* and *basR* are from the colistin dataset. Black, MG1655; Blue, MG1655 + nucleotide analogs; Red, mutator strain. **e–h** The combined rankings are plotted for ciprofloxacin or colistin datasets using either a KS test (EA-KS) or the mutation rate adjusted summation of EA scores (EA-sum). Genes previously shown to contribute towards either ciprofloxacin resistance (**e, f**) or colistin resistance (**g, h**) are shown in orange and labeled with rank. Source data are provided as a Source Data file.

resistance will likewise be enriched for functionally impactful mutations and therefore be characterized by elevated EA. Indeed, positive control genes well-known for ciprofloxacin resistance, such as *gyrA*, and *parE*, or colistin resistance, such as *basS* and *basR*, are statistically enriched (Kolmogorov–Smirnov (KS) test) for substitutions with greater EA values than the simulated random mutation background (Fig. 2a–d). The differences between mutational EA profiles of individual genes and the simulated background can be measured by the EA scores integrated over the ensemble of mutations and, in practice, can be quantified by the nonparametric KS test (EA-KS) or the sum of EA scores (EA-sum). For each antibiotic selection and mutational load, the EA-KS and EA-sum integration procedures rank every gene by the overall deviation from random of their mutations (Supplementary Data 2). For the ALE of MG1655(WT), only 24 genes have nonsynonymous mutations across the 14 replicates and 3 of

5 mutated ciprofloxacin resistance drivers are the top three genes ranked by EA-sum. For the ALE of WT + mutagen, 285 genes across 15 replicates have nonsynonymous mutations and 6 of 7 mutated ciprofloxacin drivers are in the top 30 ranked genes. For the mutator strain, 3440 genes (~80% of protein-coding genes) have at least one nonsynonymous mutation across 29 replicates and yet 7 of 8 known drivers are in the top 2% of ranked genes (Supplementary Fig. 2). Finally, to generate high-confidence testable predictions for each antibiotic selection, we utilized an aggregation-ranking method based on order statistics[28](Supplementary Data 2) that combines rankings across the three mutational regimes. This yielded the overall EA-KS and EA-sum rankings of mutated genes shown in Fig. 2e–h for each antibiotic selection condition.

In order to assess these gene rankings, we asked if they recovered the known gene drivers for ciprofloxacin and colistin resistance. Specifically, genes known to mediate chromosomally-

encoded ciprofloxacin resistance include: the antibiotic's targets (*gyrA*, *gyrB*, *parC* and *parE*), regulators of drug efflux (*acrR*, *marR* and *soxR*) and altered porin function (*ompF*)[29]. Strikingly, apart from *gyrB* and *marR*, all six remaining gene drivers are the top six ranked by EA-KS (Fig. 2e). EA-sum performed slightly better by recovering *gyrB* in the top 10 ranked genes (Fig. 2f). Less is known about the drivers of colistin resistance, but two drivers of colistin resistance in *E. coli* encode the BasR/BasS two-component system (also known as PmrA/PmrB)[1]. These genes were ranked at the top two positions by both EA methods (Fig. 2g, h). These data show EA integration can recover known antibiotic resistance genes from genome sequencing despite a noisy background dominated by hitch-hiker mutations.

### EA integration and frequency-based methods agree on top-ranked genes.

We further compared EA integration to a conventional approach that ranks genes based on the probability that their observed mutational frequency deviates from average. Genes mutated during several parallel experiments are more likely to be driving the phenotype under selection. However, longer genes have a higher chance of mutating and ignoring gene length can cause longer genes to emerge as false positives, as seen in early attempts to identify drivers from cancer exomes[30]. Thus, our frequency-based method is normalized to gene length. As with the EA ranks, the frequency-based rankings were aggregated to generate an overall rank for either the ciprofloxacin or the colistin selection experiments. The orthogonal EA integration and frequency-based method agree well (Fig. 3, Supplementary Fig. 3, Supplementary Fig. 4 and Supplementary Data 2) and correctly place most known drivers of resistance at the top of their respective gene list.

### Contribution of specific mutations to ciprofloxacin resistance.

In addition to recovering the well-known driver genes, it is possible that additional top-ranked genes contribute to antibiotic resistance. Therefore, we sought to experimentally test genes ranked highly by EA integration (here EA-KS), ranked highly by frequency statistics, or ranked highly by both methods (Fig. 3a, b). In general, the specific mutations we tested experimentally have midrange (30–70) to high (70–100) EA scores which are predicted to impact protein function (Supplementary Table 1) and are found in evolutionary important sites, as characterized by Evolutionary Trace[19] (Supplementary Fig. 5). We used P1 transduction to revert putatively adaptive mutations in the gene of interest in the evolved strains in order to test that specific variant's contribution to fitness in the genetic context in which it was found. We then conducted competition experiments between the evolved clone and it's revertant in the presence of the antibiotic that it was adapted.

As positive controls, all the tested mutations in highly-ranked known ciprofloxacin resistance driver genes, *gyrA*, *ompF*, *parE*, and *parC*[31], had a significant effect on fitness (Fig. 3c). As a negative control, a gene ranked low by all methods, *hemX*, had no detectable influence on fitness in the presence of ciprofloxacin. Competition in the presence of ciprofloxacin indicates the *rob* A70V mutation has a strong influence on fitness (Fig. 3c). This result corroborates previously observed correlations between *rob* mutations and ciprofloxacin resistance[32,33]. Closer examination of ET ranks mapped on the *rob* structure shows that, while position 70 is not predicted to be evolutionarily important, it lies directly adjacent to an evolutionarily important site (Supplementary Fig. 5d). Accounting for structural context has been shown to improve ET scores and may be able to improve future calculations of EA scores as well[34]. One other high-ranked gene, *udk*, has not been previously associated with ciprofloxacin resistance but here shows a small but statistically significant contribution towards fitness in the

evolved clone (Fig. 3c). The *udk* gene is involved in nucleotide metabolism and 29 of 30 mutations in *udk* were observed in samples with WT + mutagen condition (this includes both ciprofloxacin and colistin ALE samples). This observation suggests a role in adapting to the presence of the mutagens rather than to either ciprofloxacin or colistin. We also found that the *udk* mutation imparts a similar fitness effect when competing in rich media alone, further indicating its fitness effect acts independently of ciprofloxacin (Supplementary Fig. 6). For the ciprofloxacin-resistant cultures, none of the other tested mutations in *rstB*, *yrbG*, *sbc*C, *mutL* or *pdxY* had a measurable impact on fitness using our competition assay. It should be noted that the evolved clones with a *mutL* A271V mutation likely have an elevated mutation rate as several new mutations appeared that were unique to individual transductants (Supplementary Data 3). Although the *mutL* mutation did not show a direct contribution to fitness in the presence of ciprofloxacin, an elevated mutation rate has previously been shown to potentiate adaptation[17,35–39] and may have been helpful in adapting to the daily increase in ciprofloxacin concentrations during our selection experiment.

We also examined the minimum inhibitory concentration (MIC) of ciprofloxacin for each of the revertants. Only *gyrA* S83L and *parC* A30V revertants reduced MIC levels greater than 2-fold (Supplementary Table 2). This is likely due to the lower sensitivity of the MIC assay relative to the competition experiments. We also transferred several of the tested mutations into the wild type MG1655 background to determine if any are sufficient, on their own, to confer ciprofloxacin resistance. Only the *gyrA* S83L mutation is sufficient to confer resistance up to 0.512–1.024 μg/ml. Also, although reversion of *parC* A30V in the evolved DCM292 strain reduces resistance levels fourfold to 1.024 μg/ml, *parC* A30V in the MG1655 background does not confer any detectable resistance in our MIC assay. This agrees with previous results showing *parC* mutations only increases fluoroquinolone resistance in the presence of a *gyrA* mutation[40]. Overall, the competition assay results show that we can distinguish between mutations that specifically contribute to ciprofloxacin resistance from those that have no direct fitness effects or influence fitness independently from the presence of ciprofloxacin.

### Identification of several new driver genes for colistin resistance.

Mutations in the *basS/basR* two-component system are the most common cause of de novo colistin resistance in *E. coli*, were mutated at least once in each of the ALE samples, and are top ranked by both EA-KS and frequency (Fig. 3b). Competition experiments between the evolved strains and *basS* or *basR* wild type revertants confirmed these two genes contributed to colistin resistance in our ALE samples (Fig. 3d). In addition, in several evolved clones isolated from independent ALE cultures, reverting just the *basR* or *basS* gene to wild type was sufficient to return the evolved strain's colistin MIC similar to *E. coli* MG1655 levels (Supplementary Table 3). This indicates that *basR* and *basS* mutations are necessary for colistin resistance in the evolved clones. In addition, *basS/basR* mutations from the evolved strains were sufficient to increase the MIC of MG1655 from 0.125 to 16 μg/mL. Although the BasR/BasS 2-component system is critical to de novo colistin resistance, additional mutations in other genes may also be contributing.

Several highly ranked genes in the colistin resistance dataset have no previous association with antibiotic resistance (Fig. 3b). This could be due to a poorer understanding of how colistin resistance arises de novo. The genes *lpxD*, *asmA*, *lapB*, *ispB*, *waaQ* and *ybjX* appeared within the top 10 ranked genes of both the EA-KS and frequency-based analysis lists, suggesting potential roles as

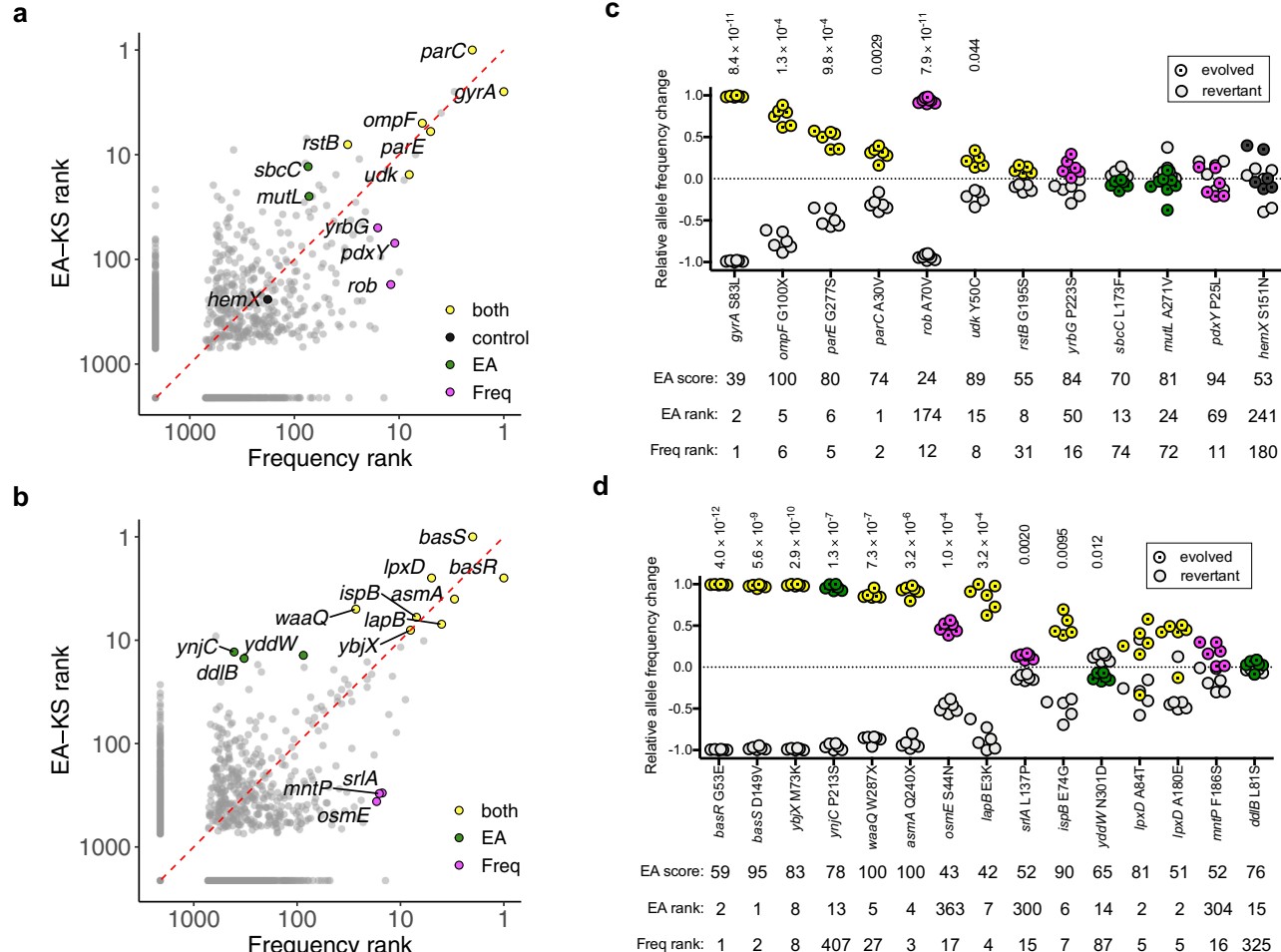

**Fig. 3 Mutations in genes highly ranked by both EA integration (EA-KS approach) and frequency analysis contribute to fitness. a, b** The combined rankings for ciprofloxacin and colistin datasets using both EA integration (EA-KS) and frequency-based analysis. Genes that were chosen for subsequent experiments are labeled with their names and colored yellow if ranked highly by both EA integrals and frequency analysis, green if EA specific, and magenta if frequency specific. The evolved strains and isogenic revertants of indicated mutations were competed in the presence of (**c**) ciprofloxacin or (**d**) colistin. In the competition assays (**c, d**), n = 4 independent biological samples for hemX. N = 5 independent biological samples for udk, rstB, and ispB. N = 8 independent biological samples for rob and mutL. All other genes were repeated with n = 6 independent biological samples. Relative allele frequency change after the competition of the evolved allele and the wild type allele are plotted. EA scores (0-100, wherein 100 is highest predicted impact) for each tested mutation are indicated along with the overall rank of each gene based on either EA-KS aggregate ranks or frequency-based aggregate ranks. One sample two-tailed t-test ($\mu_0 = 0$) with Bonferroni correction was performed for each competition assay (p values <0.05 reported above each dataset). Source data are provided as a Source Data file.

mediators of colistin resistance. Although, reversion of putatively adaptive mutations to wild type in *lpxD*, *asmA*, *lapB*, *ispB* or *waaQ*, did not alter the MIC levels of the evolved clones, reverting *ybjX* resulted in a 4-fold drop of MIC from 64 to 16 µg/mL, indicating a significant impact of this mutation towards colistin resistance in the evolved strain background (Supplementary Table 3). However, *ybjX* mutation did not increase the MIC, when introduced to MG1655 background. In the competition experiment, we detected a statistically significant change in fitness for the *asmA*, *lapB*, *ispB*, *waaQ* and *ybjX* revertants (Fig. 3d). These results indicate the evolved clones benefit from these mutations, in the presence of colistin. The only gene that did not have a confirmed effect on fitness was *lpxD*. This was surprising as this gene participates in lipid A biosynthesis[41] which is the component of LPS that becomes modified to impart colistin resistance. Also, *lpxD* mutations were also previously shown to contribute to colistin resistance in *Acinetobacter baumannii*[42]. The lack of a significant effect of two tested *lpxD* mutations upon fitness in the presence of colistin may indicate a limitation in the

sensitivity of our competition assay especially when a stronger effect mutation in a driver gene (*basS/basR*) is present. Overall, these data show that mediators of colistin resistance can be identified by focusing on genes that are highly ranked by both EA integration and frequency-based methods.

For the colistin resistant cultures, *ddlB*, *ynjC* and *yddW* are specifically picked out by EA-KS but are not highly ranked by the frequency-based method. Although the *ddlB* and *yddW* mutations do not affect the fitness of the evolved strains in the presence of colistin, reversion of *ynjC* P213S to wild type causes a striking decrease in fitness (Fig. 3d). After 24 h of competition, the evolved strain completely swept the cultures, suggesting *ynjC* P213S mutation confers a strong fitness advantage for *E. coli* under colistin treatment. However, little is known about *ynjC* other than it is predicted to be a subunit of an inner membrane transport complex[43].

The genes *mntP*, *srlA* and *osmE* were specifically identified by frequency-based analysis but not EA integration (Fig. 3b). The *osmE* and *srlA* genes have statistically significant contributions

towards fitness (Fig. 3d), although the contribution from *srlA* is minimal. For *osmE* and *srlA*, the protein family alignments had poor sequence diversity making EA predictions less reliable. Increased availability of sequence homologs as more sequences become available should help improve EA predictions made on these genes.

Overall, these results suggest that genes that are ranked highly by both EA-KS and frequency-based method are more likely to contribute to fitness. In addition, some genes with fitness-enhancing mutations can be specifically predicted by EA but not frequency-based analysis (e.g. *ynjC* P213S) or vice versa (e.g. *rob* A70V and *osmE* S44N). Also, a caveat with any of the mutations that show no fitness effects is that they may have an indirect role on resistance by increasing adaptability, which may require serial passaging over multiple days in order to be detected by the competition assay.

**EA integration applied to environmental and clinical datasets**. We next applied EA integration to publicly available *E. coli* whole-genome sequencing datasets of clinical and environmental isolates that are either ciprofloxacin or colistin resistant[3,44]. In contrast to an experimental evolution design, such clinical and environmental isolates rarely have matched samples prior to antibiotic treatment and have several additional complex variables including patient or environmental or strain heterogeneity, potential exposure to multiple antibiotics, selection for commensalism, competition within unique microbiomes, differing isolation and sequencing methods, and, especially, the potential for horizontal gene transfer. Existing methods to track antimicrobial resistance typically depend upon established antimicrobial resistance genes and associated genetic elements but do not address the emergence of de novo resistance from heretofore uncharacterized mechanisms[45]. With this challenge in mind, we applied both EA integration approaches to 62 ciprofloxacin-resistant *E. coli* isolates obtained from multiple environments and 146 colistin-resistant genomes from clinical isolates lacking the plasmid-borne *mcr* colistin resistance genes. Because clinical and environmental strains have undergone negative selections during their evolutionary history, the mutation background should deviate from randomly simulated mutations and needs to be reestablished. In order to address this, we generated background distributions using unique nonsynonymous mutations from ciprofloxacin and colistin sensitive environmental/clinical strains. Their EA distributions are independent but nearly identical (Supplementary Fig. 1b, c) and fit decaying exponentials well, with Pearson $R^2$ of 0.98. Although environmental/clinical strains have abundant mutations, they are strongly biased to low EA scores (average ~21) when compared to randomly simulated nonsynonymous mutations (average ~42). This strong bias is consistent with negative selection leading to most of the observed mutations being nearly neutral with only a small fraction contributing to functional differences. Compared to the frequency-based approach, both EA-KS and EA-sum are better at separating several known drivers of de novo resistance from a large background of variation in the other genes (Fig. 4, Supplementary Data 4). Specifically, EA-KS and EA-sum place the ciprofloxacin drivers in the top 1% and the colistin drivers in the top 0.1% of genes mutated in the resistant isolates. In contrast, the frequency-based approach places the four main drivers of ciprofloxacin resistance in the top 10% alongside ~280 other genes. Likewise, with colistin resistance, *basS* and *basR* were in the top 6% alongside ~230 other genes. In contrast to the well-controlled experimental evolution dataset, in which EA integration and frequency-based analysis largely reinforce each other, EA integration is superior at separating known drivers from a large

background of strain-to-strain variation. The newly identified ciprofloxacin and colistin resistance drivers in the ALE datasets were not recovered in the clinical and environmental dataset. But, the top-ranked genes by EA integral cluster significantly in Stringdb (Supplementary Table 4)[46], indicating the protein products of the highly ranked genes associate with each other physically or are functionally related. For instance, *mukB* was ranked 7th by EA-KS in the ciprofloxacin environmental dataset. It was reported that *mukB* interacts with *parC*, a known ciprofloxacin resistance driver, and stimulates the superhelical DNA relaxation activity of wild-type Topo IV[47]. Another top-ranked gene, *arnT* (ranked 23rd by EA-sum), in the colistin clinical dataset was reported to be activated in polymyxin/colistin-resistant *E. coli*[48,49].

**Evaluation of method robustness**. Our results showed that the predictions of phenotype driver genes were improved over the frequency-based method, especially in the clinical/environmental datasets, when we weight the mutational impact of each amino acid substitution with its EA scores. We further examined if EA can be substituted with SIFT scores, which is commonly used to predict the impact of amino acid substitutions on protein function[50–52]. The EA scores computed for *E. coli* K12 MG1655 show a good correlation with SIFT scores (Supplementary Fig. 7), with median gene-level Pearson and Spearman correlations of −0.54 and −0.73, respectively. When SIFT scores are substituted for EA scores in the integration (KS test and EA-sum), the experimentally validated genes ranked similarly with the exception of *ynjC* which was not recovered in colistin ALE datasets (Supplementary Fig. 8). These findings suggest that the integration of mutational impact scores is a robust approach to predicting phenotype driver genes, as EA scores can be substituted by other prediction methods although this entails a slight loss of sensitivity (*ynjC*).

The experimental evolution experiments were conducted on a large number of replicates (from 14 to 31 depending on the conditions) in 3 different mutational loads per antibiotic tested. However, this large number of replicates may not be necessary to detect genes under selection. In order to address this, down-sampling analyses were performed to ascertain if similar results can be achieved with fewer samples. We first examined the number of antibiotic-resistant drivers (known and newly identified) that were mutated in the subsamples (Supplementary Fig. 9). As expected, more drivers were mutated in the WT + mutagen and mutator conditions (8–9 for each antibiotic) compared to WT (5-6). Approximately 10-12 samples are required in order to observe at least one mutation in each phenotypic driver. We further performed EA and frequency analyses on the subsamples and evaluated how the antibiotic resistance drivers were ranked using different sample sizes. Our methods were able to recover most drivers in the ALE datasets with fewer than 10 samples (Supplementary Figs. 10–12). In the more complicated clinical/environmental datasets, EA integral was able to reproduce similar results as the full dataset with around 40 strains but still performed well with just 10 samples (Fig. 4e, f). These findings further suggested our approach is generally robust.

## Discussion

The coupling between genotype and phenotype determines major aspects of biology but remains cryptic. A global approach describes it through fitness landscapes, which may be mapped experimentally by deep mutational scans coupled to assays[53–57]. Such scans, however, require exhaustive mutagenesis and a complete battery of relevant biological assays, which are impractical for more than a

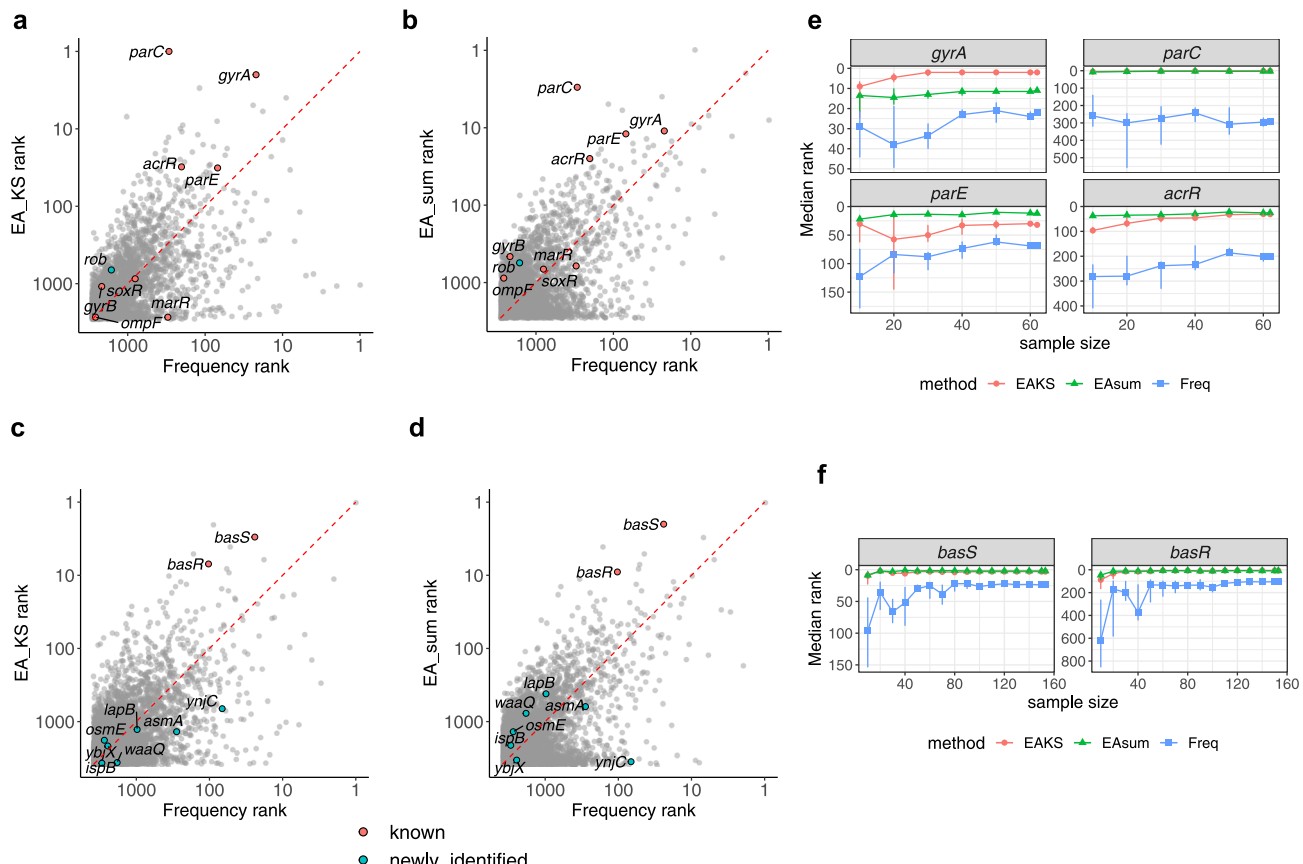

**Fig. 4 Comparisons of EA integration with frequency-based analysis in clinical/environmental datasets.** Frequency analysis and EA integration ranks (EA-KS and EA-sum) of the mutated genes in ciprofloxacin resistant (**a**, **b**) and colistin resistant (**c**, **d**) clinical/environmental strains. Known drivers (orange) and genes with mutations that contribute to fitness in the presence of antibiotic (cyan) are highlighted. Median rank of ciprofloxacin resistance driver gene ranks (**e**) and colistin resistance driver gene ranks (**f**) in down-sampling analysis wherein random subsets of the samples were chosen for analysis by EA-KS (red/circles), EA-sum (green/triangles) or frequency (blue/squares). Interquartile range is displayed as error bars from 10 independent random draws at the indicated sample size. Source data are provided as a Source Data file.

few genes. As a result, experimental descriptions of fitness landscapes are incomplete and beyond reach. The evolutionary action addresses these limitations by substituting the evolutionary history of all genes accessible from sequence databases instead of the required laboratory experiments. Specifically, EA theory develops a calculus approach that uses these evolutionary histories to exhaustively measure mutational displacements in a fitness landscape in response to selection pressures. This calculus is based on four hypotheses. First, that comparing sequence homologs across a protein family to tally coding variations at every sequence position is similar to carrying mutations in a laboratory. Second, that coupling these mutations to the depth of associated evolutionary divergence between homologs will estimate the likelihood of acquiring distinctive functions upon mutation, and is similar to a battery of functional and relevant assays. Third, that multiplying the size of a mutation at a position by the functional sensitivity of that position, as in Eq. (2) in Methods, follows a fundamental concept of calculus to quantify the impact of a perturbation, which here is the fitness effect of a mutation. Fourth, that summing the fitness effects of all mutations over a population to compare and contrast, for each gene, deviations from random, is an integration of Eq. (2) that will recover outlier genes traveling nonrandomly in the fitness landscape, and thus identify genes that drive adaptation to a selection condition in ALE. Propositions one, two and three have already been extensively tested and validated. For example, EA scores correlate with mutational scanning experimental data[18] and predict the harmful effect of mutations in diverse tests (CAGI) and

applications[20,58–61], reflecting its embodiment of a large amount of evolutionary mutational scanning data. Proposition four, however, has not been tested directly, although prior analysis of EA differences over the entire population of patients have been linked to genes driving human phenotypes, such as, parathyroid cancer[58], modification of APOE2/3-based Alzheimer's disease risk[21], and the depth of autism[22]. These studies, however, do not directly test proposition four since they do not explicitly carry out integration by summing the EA scores. Moreover, they focus on complex human diseases with environmental and social confounding etiologic factors that cannot be controlled for fully. Finally, computational association of specific genes to complex human disease are useful but difficult to validate with certainty. For these reasons, this study focuses on the validation of proposition four by testing the direct causal link between antibiotic resistance phenotypes and genes biased for mutations with elevated EA scores, in a simple, well-controlled and easily testable evolutionary selection system. The results establish new genes for colistin resistance that are not previously associated with an antibiotic phenotype (Fig. 5). Since colistin is a last resort antibiotic, a better understanding of the mechanism of this resistance is significant medically. This application also strengthens the calculus of fitness landscapes embodied in the EA theory.

Although the cationic region of colistin is known to interact with the anionic lipid A subunit of lipopolysaccharide (LPS) at the bacterial outer membrane[62], the details of the downstream killing mechanism of colistin are still unknown[1]. The current

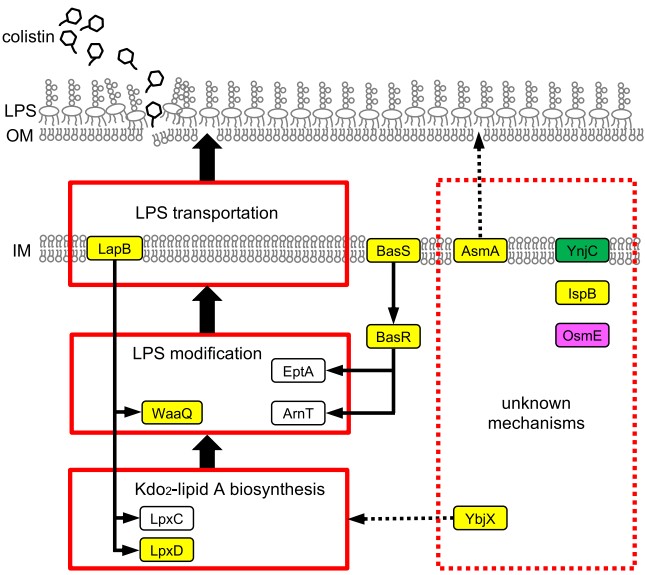

**Fig. 5 Overview of genes under selection in the cultures experimentally evolved to be colistin resistant.** Genes highly ranked by both EA Integral and frequency-based analysis are colored yellow; EA-specific genes, green; frequency-specific genes, magenta.

model is that colistin and related polymyxins first disrupt the outer membrane by binding lipid A to displace $Mg^{2+}$ and $Ca^{2+}$ and then, cross the permeabilized outer membrane to bind and disrupt the phospholipid inner membrane[63]. Recently it has been shown that active LPS synthesis is required for effective colistin-mediated killing of *Pseudomonas aeruginosa* suggesting that alterations in LPS synthesis inhibit the antibiotic's activity[64]. Several of the genes that were newly identified here are related to LPS and may be altering the structure of this colistin target directly or modifying the LPS synthesis pathway. The *lapB* and *waaQ* gene products are involved in LPS synthesis[65–67], while *asmA* null mutants were previously shown to lower the LPS levels in the outer membrane[68]. Detailed studies of *lapB* found it to be essential for growth in normal laboratory conditions (37°, rich media) and suppressors of this essentiality include mutants of *waaQ* and *lpxD*[66,67]. In *Salmonella enterica* serovar Typhimurium, the *ybjX* homolog has been shown to partially revert a growth defect that was caused by a mutation in the biosynthesis of an LPS precursor, lipid A[69]. In addition, *ybjX* was shown to be involved in the development of resistance in *Salmonella enterica* serovar Enteritidis to polymyxin B, an antibiotic structurally and functionally related to colistin[70].

Mutations in the *ispB* and *ynjC* genes contribute to fitness in the presence of colistin but these genes have no previous association with LPS synthesis or modification. The *ynjC* gene would not have been linked to colistin resistance using a standard frequency-based approach due to the low mutation count (1 in WT + mutagen and 6 in mutator) (Supplementary Fig. 4 and 13). However, those mutations were predicted to be impactful by EA. Although the mechanism by which *ispB* and *ynjC* are linked to colistin resistance is unknown, each of these genes is predicted to encode a membrane-bound protein and may affect function of the inner membrane, the outer membrane or LPS itself. IspB synthesizes octaprenyl diphosphate, a precursor of membrane-associated ubiquinone-8 which has been shown to be a membrane-stabilizing osmoprotectant[71,72]. YnjC is a predicted inner membrane subunit of a putative ATP-dependent transporter complex[43] but how this protein confers colistin resistance is unknown. Although there are no direct studies regarding the substrate of *ynjC*, this gene is annotated as encoding a membrane component of a thiamine

ABC transporter complex that transports sulfate, thiosulfate or thiamine[43]. However, deletion of *ynjC* reduced intake of one of the two tested cationic dyes in a screening assay[73] and thereby *ynjC* mutations may influence uptake of colistin as it is also a cationic compound. The unexpected contribution of these genes towards colistin resistance warrants further investigation and may provide a better understanding of the relationship between LPS assembly, the osmotic stress response and colistin resistance.

In the context of ciprofloxacin resistance, which is much better studied, EA integration and frequency-based methods reinforce each other to arrive at high-confidence identification of the known drivers in the ALE experiments. However, this is in contrast to EA's performance on the clinical and environmental isolates, in which EA performs substantially better than the typical frequency-based approaches in discriminating known drivers from a large background of variation. This difference may be related to the clinical and environmental samples lacking matched references before antibiotic treatment. Without ideal reference sequences, any preexisting founder mutations are indistinguishable from newly acquired resistance mutations and are given the same statistical weight by frequency-based analysis. EA scores can mitigate this effect by predicting the phenotypic impact of a mutation and thereby place emphasis on mutations that are more likely to change protein function.

The genes identified in this study stand out from the background by having mutations with a larger than expected tally of EA scores. Mathematically, this difference from background is a definite integral, which sums the EA scores observed and subtracts EA scores of the expected background, to find genes for which this difference is significant. Another interesting observation is the exponential form of the background distribution of EA scores, which reflects that, under a steady state, mutations are biased to lower fitness effect. This is consistent with evolutionary models of the fitness effect distribution as well as with physical models of the fitness landscape. The fitness landscape is an energy potential and the distribution of mutations falls off as an exponential Boltzmann function of their energy[27]. We can reconcile this physical view of fitness landscape with EA by describing explicitly the genotype to phenotype function as an evolutionary potential function. The slope of the landscape, or gradient, is then a force. The product that defines EA is then a force times a displacement in genotype, which is the fitness energy cost of the mutation as it moves a genome in the fitness landscape. In this view, when a population is maladapted to a new environment, such as exposure to antibiotics, it needs to modify its genome to reach a different fitness location. Low impact, low energy mutations are not sufficient to reach a higher location of the landscape. Organisms with specific high impact/high energy mutations in genes functionally related to the new environment are selected because these mutations have sufficient energy to allow the organism to take larger steps as it climbs the new fitness peak. These few high-energy mutational steps in genes relevant to the new phenotype are in contrast to the many smaller mutational steps that occur in the genes unrelated to the selected phenotype. EA theory approximates these energies and their distribution in light of the rich record of evolutionary history. As shown here for ciprofloxacin and colistin resistance, upon integration, gene outliers should be closely related with the selection pressures or phenotypes specific to the population under study.

## Methods

**The evolutionary action theory and extension to identify driver genes.** The evolutionary action models the genotype-phenotype relationship formally as:

$$f(\gamma) = \varphi \qquad (1)$$

Here $f$ is the evolutionary potential. It maps any genotype $\gamma$ to its complete set of biological capabilities $\varphi$. Thus, given a specific environment, $\varphi$ embodies the overall fitness of an organism with genotype $\gamma$, and $f$ described the fitness landscape in that

environment. Assuming differentiability, a mutation is a small change in the genotype $\triangle\gamma$, which calculus suggests should trigger a fitness response $\triangle\varphi$ in proportion to the magnitude of $\triangle\gamma$ and to the gradient of the potential $\nabla f$, so that we expect:

$$\triangle\varphi = \nabla f \cdot \triangle\gamma \qquad (2)$$

By analogy to physical phenomena, the gradient of the potential $f$ is an Evolutionary Force that describes the local slope of the fitness landscape. Its product with $\triangle\gamma$ is $\triangle\varphi$, which we call the evolutionary action (EA) of the mutation on fitness[18], and, being a force ($\nabla f$) times a displacement ($\triangle\gamma$), it represents the work done by the mutation as it moves the genome in the fitness landscape. In practice, both of these terms in the right-hand side of Eq. (2) can be approximated from sequence data for coding variants. For a single point coding mutation from amino acid $X$ to $Y$ at a sequence position $i$, $\nabla f$ is approximated by the evolutionary sensitivity of position $i$, given by the Evolutionary Trace (ET) algorithm[19]. And the magnitude of the substitution $\triangle\gamma$ is approximated by amino acids substitutions log odds tables. This means that Eq. (2) is computable for coding mutations, even if $f$ itself in Eq. (1) remains unknown.

To recover the genotype-phenotype relationship of Eq. (1), the fundamental theorem of calculus states that integrating Eq. (2) should suffice, since it is the anti-derivative operation. Therefore, we wish to compute:

$$\int_{C_j} f'(\gamma)d\gamma = \varphi\big(C_j\big) \qquad (3)$$

where the integral is over all mutations in all genes from the sequenced population of E. coli $C_j$ with the antibiotic resistance ($j$) of interest. In practice, (3) is evaluated numerically as a sum:

$$\overset{\text{ALE mutations}}{\underset{C_j}{\sum}} f'(\gamma) \cdot d\gamma - \overset{\text{random mutations}}{\underset{C_j}{\sum}} f'(\gamma) \cdot d\gamma = \varphi\big(C_j\big) \qquad (4)$$

where the subtracted second term is an arbitrary integration constant, chosen here to zero out random mutations in bystander genes. Next, since genes are the functional units of genotype, we can rearrange the summation (4) to compute it gene by gene, for each gene $k$:

$$\underset{C_j}{\sum} f'_k(\gamma) \cdot d\gamma - \overset{\text{background mutations}}{\underset{C_j}{\sum}} f'_k(\gamma) \cdot d\gamma = \varphi_k\big(C_j\big) \qquad (5)$$

An individual gene $k$ thus makes no contribution when its mutations in $C_j$ are no different from what is expected from random chance. However, if gene $k$ is a resistance driver gene, it should have a non-zero value in (5) and contribute to $\varphi(C_j)$. This enables us to detect genes with mutations associated with the trait $C_j$. Below, we compute this EA integral on the mutations in strains grown under selection pressure and test whether the integral recovers known driver genes of antibiotic resistance and also whether it identifies new genes not previously associated with resistance.

It should be noted that, in practice, the calculation of EA scores is performed separately for each gene and currently no weights are given for genes that are more conserved across species. For instance, a predicted low impact mutation in an essential gene like gyrA may influence fitness more than a predicted high impact mutation in a non-essential gene like pdxY.

**Culture medium.** Lennox lysogeny broth (LB, BD Biosciences, Franklin Lakes, NJ) and plates were used for routine propagation. M9 glucose minimal media was used in ALE experiments and consisted of 0.1 mM CaCl$_2$, 1× M9 salts, 2 mM MgSO$_4$, 0.2% glucose (Pfaltz & Bauer, Waterbury, CT) and 10 μg/ml thiamine (Acros Organics, Geel, BE). The 1× M9 salts were diluted from 5× M9 salts which was made by dissolving, into 1 L of deionized water, the following: 64 g Na$_2$H-PO$_4$−7H$_2$O (Santa Cruz Biotech, Dallas, TX), 15 g KH$_2$PO$_4$ (J.T. Baker, Phillipsburg, NJ), 2.5 g NaCl (MilliporeSigma, Burlington, MA) and 5 g NH$_4$Cl (J.T. Baker). The mutagenic synergy of nucleotide analogs[23] was used to elevate mutation levels in the WT + mutagen ALE experiments by including 250 μg/ml 2-aminopurine (Alfa Aesar, Ward Hill, MA) and 2.5 μg/ml zebularine (Enzo Life Sciences, Farmingdale, NY) in the M9 glucose minimal media.

**ALE.** Overnight cultures grown in LB were inoculated from isolated colonies on LB plates. These overnight cultures were diluted 1:100 into 50 ml conical tubes containing 5 ml M9 glucose minimal media containing either a minimal inhibitory concentration of antibiotic (the challenge tube) or ½ the minimal inhibitory concentration (the maintenance tube). A portion of the overnight starter culture grown in LB was saved as a day 0 control for sequencing and identification of founder mutations. After 24 h of growth at 37 °C with shaking at ~300 r.p.m., the optical density at 600 nm (OD$_{600}$) was determined. If the challenge tube reached an OD$_{600}$ > 0.4, then culture from this tube was used at a 1:100 dilution to start the next round wherein the antibiotic concentration was doubled. If the challenge tube's OD$_{600}$ was less than 0.4 and the maintenance tube's OD$_{600}$ was greater than 0.4, then culture from the maintenance tube was used at a 1:100 dilution to start the next round wherein the antibiotic concentrations remained the same. If neither

tube had an OD$_{600}$ > 0.4, the cultures were returned to shake for another 24 hrs. In this way, less successful cultures had more time to adapt.

**Whole-genome sequencing.** After an evolution experiment, 4 mL of evolved cells from tubes that would be sequenced were pelleted. Genomic DNA was extracted from the cells using the QIAGEN (Venlo, NL) Dneasy Blood & Tissue Kit. The purified DNA was then quantified with the Promega (Madison, WI) QuantiFluor dsDNA System and adjusted to 0.2 ng/μL. The sequencing libraries were prepared with Illumina (San Diego, CA) Nextera XT DNA Library Prep Kit and sequenced with Illumina Miseq v2 or v3. All sequencing data are available at NCBI SRA PRJNA543834.

**Calculation of EA scores.** EA scores were calculated[18]. Homologs for all protein-encoding genes of the E. coli MG1655 reference genome U000096.3 were obtained from the NCBI nr, UniRef100 and UniRef90 databases[74]. The homologs for each protein gene are retrieved from different genomes because different proteins have different evolutionary rates throughout evolutionary history. From each database, up to 5000 sequences were retrieved using an e-value cut-off of $10^{-5}$ and a minimum of 30% sequence identity. The recovered sequences from each database were aligned using MUSCLE[75] to generate three separate multiple sequence alignments for each protein. Next, the rvET method was used to rank amino acids by evolutionary importance[76–78]. The rvET scores from the three alignments were then averaged. The second parameter of EA, the log-odds of amino acid substitution, was computed following BLOSUM methodology[79] except separate odds were calculated according to deciles of rvET score. In other words, positions with rvET scores 1-10 had a separate log-odds substitution matrix calculated from positions with rvET scores of 11–20 and so on up to the 91–100 decile. Odds were calculated based on over 67,000 multiple sequence alignments and rvET scores generated from proteins deposited in the Protein Data Bank.

**Analyzing ALE data.** Sequencing reads were mapped to E. coli K-12 MG1655 reference genome and variants were called by breseq version 0.31.0 using the "–no-junction-prediction" option to ignore reads that partially map to the reference and "-p" option to estimate allele frequency in the population[80]. Mutations with allele frequency smaller than 0.1 were removed from further analysis. A recalculated EA score (ranges from 0 to 100) was assigned to each missense mutation. Nonsense mutations were arbitrarily assigned with EA = 100. All mutations that appeared in the evolved strains were pooled together according to the condition under which they were selected (level of mutagenesis and antibiotic) and had founder mutations removed to yield 6 datasets. Each dataset was processed individually. 10 random MG1655 genomes, each with 10,000 point mutations, were generated using breseq. Coding mutations were called and tallied. A total of 63,507 coding mutations have EA scores assigned, and they were used as randomly simulated background. In order to identify genes that acquired a significantly different distribution of mutations during the evolution experiments, EA integration was approximated with a one-sided Kolmogorov–Smirnov test (EA-KS) or the sum of EA scores (EA-sum). EA-KS was performed between the EA distribution of each gene against the in silico generated random EA distribution background. The p-values for the KS tests were used to rank the genes. EA-sum for a given gene $i$ with $m_i$ mutations in the samples is calculated with:

$$\text{EAsum}_i = \sum_{j=1}^{m_i} \text{EA}_{ij} - E\big(\text{EAsum}_i\big)$$

$$= \sum_{j=1}^{m_i} \text{EA}_{ij} - E\big(m_i\big) \times E(\text{EA}_{ij})$$

$$= \sum_{j=1}^{m_i} \text{EA}_{ij} - \frac{\sum_1^g m_i}{\sum_1^g l_i} \times l_i \times E(\text{EA}_{ij})$$

$$= \sum_{j=1}^{m_i} \text{EA}_{ij} - \frac{\sum_1^g m_i}{\sum_1^g l_i} \times l_i \times \text{EA(random)} \qquad (6)$$

Where EA$_{ij}$ is the EA score of the $j^{th}$ mutation in gene $i$, $l$ is the gene length, $m$ is the mutation count, $g$ is the number of genes in MG1655, EA(random) is the average EA scores of random mutations and $E$ is the expected value. Frequency-based analyses were performed based on the assumption that the probability of x mutations occurring in a protein with given length $l$, follows a Poisson distribution with $\lambda = l \times m$, where m is the average mutation rate in each dataset. The frequency p-value for each gene was calculated by $p = P[X \geq x]$. One EA integral-based and one frequency-based gene list was generated for each dataset. An aggregate ranking algorithm (R package: RobustRankAggreg version 1.1)[28] based on order statistics was used to combine three EA integral lists or frequency lists from the same antibiotics condition into one single list (Supplementary Data 2).

**Evolutionary trace on the top ranked genes or their homologs.** The evolutionary trace of gyrA (PDB: 1AB4), parC (PDB: 1ZVU), parE (PDB: 1S16), rob (PDB: 1D5Y), udk (homolog from Thermus thermophilus, PDB: 3W8R), basR (homolog from Klebsiella pneumoniae JM45, PDB: 4S04) and ispB (PDB: 3WJK) were calculated using the UET web server[81]. The ET scores were color mapped to the protein structures with PyMOL 2.5.2 (Schrodinger, LLC)[82].

**Reverting mutations in evolved strains**. All strains used in this study are listed in Supplementary Data 3. The genome library collection of 94 intergenic kanamycin resistance cassette insertions in *E. coli* MG1655 (i-Deconvoluter) was previously generated by Nehring et al.[83]. P1 lysate was generated with a strain from i-Deconvoluter library that has the kanamycin marker nearest to the gene of interest. An overnight culture of the selected i-Deconvoluter strain was diluted 1:50 in 10 mL of phage broth (LB + 5 mM CaCl$_2$ + 0.2% glucose) and incubated at 37 °C for 2 hours. 10 μL of 2-Mercaptoethanol was added to the cells. After 30 min of incubation, the cells were spun down, resuspended in 1 mL of phage broth and then infected with 400 μL of P1*vir*. After the mixture was shaken at 37 °C for 20 min, 7.5 mL of molten agar was added and distributed onto 3 bottom agar plates. The plates were incubated at 37 °C overnight. The top agar was then transferred to Beckman tube, treated with 300 μL of chloroform and incubated at room temperature for 30 min. Debris were pelleted (10000 rpm, 10 min, 4 °C) and supernatant (P1 stocks) were collected and stored at 4 °C for future use.

An overnight culture of the recipient cell was diluted 1:50 into 2 mL of LB. After 2 h of growth at 37 °C, the OD$_{600}$ nm was measured. Cells were pelleted (7000 rpm, 2 min) and adjusted to OD = 0.45 with LB + 5 mM CaCl$_2$ + 10 mM MgSO$_4$. 0.1 mL of phage was mixed with 1.3 mL of cells and incubated at 37 °C for 20 min. The cells were then pelleted (7000 rpm, 2 min) and resuspended in 150 μL LB with 100 mM sodium citrate (MilliporeSigma). After 1 hr of incubation at 37 °C, the transductants were plated on LB plate with 100 mM sodium citrate and 40 μg/mL of kanamycin (Fisher Bioreagents, Waltham, MA) to select for transductants that received the kanamycin resistance cassette and ~100 kb of wild type genomic DNA flanking the cassette. Individual clones were sequenced using primers listed in Supplementary Data 5. to detect the presence of the wild type allele.

**Generating target mutations in MG1655 background**. We used 2 different approaches to generate target mutations in MG1655 background. The specific method used for each strain is listed in "Supplementary Data 3. Strains used in this study". For the P1 transduction approach, P1 lysate was generated with a strain from i-Deconvoluter library[83] that has the kanamycin marker nearest to the gene of interest. Then the lysate was P1 transduced into the evolved strain that contains the target mutation. Individual clones of the transductants were screened for the target mutation. This strain was then used to generate P1 lysate, and further transduced into MG1655. Individual clones were sequenced for the target mutation. To generate the other strains, we used CRISPR-FRT[84]. In brief, P1 lysate was generated with a strain from KEIO KO library that has the deletion of the gene of interest, and then this gene KO was P1 transduced into MG1655. CRISPR-FRT plasmids were then transformed into the strain through electroporation. Overnight culture was diluted 1:50 in LB + 0.2% arabinose and grown at 32 °C for ~3 hr. Rescue DNA was generated by PCR using the evolved strain that contains the target mutation with primers binding at around ±200 bp of target gene, and then the purified product was transformed into the cells and plated on LB plates containing aTc to induce expression of Cas9/sgRNA-FRT and antibiotics to select for plasmids. Single clones were screened for kanamycin sensitivity, and then PCR products sequenced to check for presence of the target mutation. Strains will be made available upon request to the corresponding authors.

**Minimum inhibitory concentration (MIC)**. The MIC of ciprofloxacin or colistin on selected *E. coli* strains were tested with microbroth dilution in LB[85]. In brief, overnight cultures were diluted 1:100 in LB and incubated for 2 hour. Cells were adjusted to OD = 0.1, and further diluted 1:100 in LB. 50 μL of cells were mixed with 50 μL of LB with colistin in 96 well plates (Greiner Bio-One, Frickenhausen, DE) to reach final antibiotic concentrations of 65.54, 32.77, 16.38, 8.192, 4.096, 2.048, 1.024, 0.512, 0.256, 0.128, 0.064, 0.032, 0.016, 0.008, and 0.004 μg/mL for colistin. After overnight incubation, the minimal antibiotic concentration that can inhibit *E. coli* growth was recorded. Each strain was tested at least 3 times.

**Competition assay**. Overnight cultures of two competing strains were mixed at a 1:1 ratio. The mixed culture was diluted 1:100 in LB and grown at 37 °C for 24 hours with or without the presence of antibiotics, which recapitulate the adaptive evolution conditions where cultures are allowed to enter stationary phase during overnight growth. The original and overnight mixtures were used as templates in PCR (Q5 NEB, Ipswich, MA) reactions to amplify the gene of interest. The PCR product was sent for Sanger sequencing (Baylor College of Medicine sequencing core, Houston, TX or Genewiz, South Plainfield, NJ). The obtained electropherograms were analyzed with R according to the method described by Carr et al[86]. for quantification of the allele frequency of the two variants. The relative allele frequency change for a given mutation was calculated by:

$$\text{relative AF change} = \frac{\text{AF}_{24} - \text{AF}_0}{1 - \text{AF}_0} \quad (7)$$

where AF$_0$ is the AF of the mutation in the original mixture, AF$_{24}$ is the AF of the mutation after 24 h of competition. Each mutation was tested at least 4 times using biological replicates. One sample t-tests ($\mu_0 = 0$) were performed. Mutations with adjusted p-values (Bonferroni correction) smaller than 0.05 were considered as having a fitness effect.

**Clinical/environmental data**. The ciprofloxacin clinical data set was downloaded from European Nucleotide Archive PRJEB23294[44]. It consists of 192 ciprofloxacin sensitive and 63 resistant environmental *E. coli* isolates. The whole-genome sequencing data was assembled using SPAdes genome assembler v3.13.0[87]. ORFs were predicted through GeneMarkS-2[88]. Predicted partial genes were removed from downstream analysis.

The colistin clinical data set was obtained from SRA BioProject PRJEB28020[3], which consist of 146 colistin-resistant clinical isolates that are caused by chromosomal mutations. A set of 313 reference *E. coli* genomes that were used by Bourrel et al. were used here as sensitive isolates. The annotated protein sequences were downloaded from the database and directly used for downstream analysis.

**Modelling of clinical datasets**. Nonsynonymous mutations were called against *E. coli* K-12 MG1655 proteome with an internal R script. If a protein sequence could not be aligned to MG1655, it is removed from future analysis. An EA score was then assigned to each nonsynonymous mutation.

The clinical/environmental data sets were first analyzed in a similar approach as our adaptive laboratory evolution data. Mutations from the sensitive isolates were removed from the resistant isolates. Mutations that occurred in the sensitive isolates were also used as background mutations. EA-KS was performed between the EA scores of all the mutations in a gene against the background. The genes were ranked based on p-values. For a given gene that has m mutations in the resistance strains, and n mutations in the sensitive strains, EA-sum was calculated with $\text{EAsum} = \frac{\sum_1^m \text{EA}_i}{S_{\text{resistant}}} - \frac{\sum_1^n \text{EA}_j}{S_{\text{sensitive}}}$ (8), where $S_{\text{resistant}}$ and $S_{\text{sensitive}}$ are the number of resistance and sensitive strains in the dataset. Genes were then ranked based on EA-sum values. Frequency-based analyses were performed based on the assumption that the probability of x mutations occurring in a protein with given length l, follows a Poisson distribution with $\lambda = l \times m$, where m is the average mutation rate in each dataset. The frequency p-value for each gene was calculated by $p = P[X \geq x]$.

**Downsampling analyses**. For the ALE data, subsamples with different sizes (from 1 to maximum sample size at interval of 1) were randomly drawn from the full datasets. The number of phenotypic drivers (ciprofloxacin: *gyrA*, *gyrB*, *marR*, *acrR*, *parC*, *parE*, *soxR*, *ompF* and *rob*; colistin: *basS*, *basR*, *lapB*, *waaQ*, *asmA*, *ybjX*, *ispB*, *ynjC*, and *osmE*) that have at least one mutation in the sub-sample were determined. Each subsample size was repeated 10 times. Each subsample was then analyzed with EA integration and frequency-based methods. Note that the downsampling was performed on the evolved strains only. All day 0 strains were used in the analyses. Gene ranks for the phenotypic drivers were determined. Median and interquartile range for each phenotypic driver was computed for each subsample size. If a phenotypic driver was not mutated in a subsample, that subsample was ignored when computing the median gene rank and interquartile range.

The same concept was applied to the clinical/environmental dataset. Subsamples were drawn at sizes 10−60 at an interval of 10 for ciprofloxacin dataset (maximum sample count of 62) and 10−150 at an interval of 10 for colistin dataset (maximum sample count of 160). GyrA, parC, parE and acrR were used as drivers for ciprofloxacin, while basS and basR were used as drivers for colistin, because other driver genes were not picked up by any methods in the clinical dataset. Note that the downsampling was performed on the resistant strains only. All the sensitive strains were used in the analyses.

**Reporting summary**. Further information on research design is available in the Nature Research Reporting Summary linked to this article.

## Data availability

The raw sequencing data generated in this study have been deposited in the Sequence Read Archive database under SRA accession code PRJNA543834. The *E. coli* K12 MG1655 reference sequence U00096.3 was used to map reads and make SNP calls. Protein Data Bank entries for *gyrA* (PDB: 1AB4), *parC* (PDB: 1ZVU), *parE* (PDB: 1S16), *rob* (PDB: 1D5Y), *udk* (homolog from *Thermus thermophilus*, PDB: 3W8R), *basR* (homolog from *Klebsiella pneumoniae* JM45, PDB: 4S04) and *ispB* (PDB: 3WJK) were used to map ET scores on structures. The data generated in this study are provided in the Supplementary Information/Source Data file. EA scores for all mutations in *E. coli* MG1655 and our proposed methods are available on our website (http://bioheat. lichtargelab.org). Our website can intake lab evolved *E. coli* sequencing data and identify phenotype driven genes using methods described here. In addition, EA scores for specific mutations in *E. coli* MG1655 can be queried through this website. The data and analyses presented here can be viewed and accessed from the interactive web page (http://bioheat. lichtargelab.org) and in the online supplementary material.

## Code availability

The code used in the analyses presented here can be viewed and accessed at https:// github.com/LichtargeLab/EA_antibiotics_resistance[89].

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

## Acknowledgements

This work has been supported by the Office of the Director of National Intelligence (ODNI), Intelligence Advanced Research Projects Activity (IARPA) under BAA-17-01 [contract #2019-19071900001 to O.L.]. The views and conclusions contained herein are those of the authors and should not be interpreted as necessarily representing the official policies, either expressed or implied, of ODNI, IARPA, or the U.S. Government. The U.S. Government is authorized to reproduce and distribute reprints for governmental purposes notwithstanding any copyright annotation therein. This work was also supported by the National Institutes of Health (R35-GM122598, DP1-AG072751 and R01-CA250905 to S.M.R.; DP1-AI152073 to C.H.; R01-GM066099, R01-AG074009, U01-AG068214 and R01-AG061105 to O.L.) and the National Science Foundation (DBI-2032904 to O.L.). We thank the Baylor College of Medicine sequencing core and Harikumar Govindarajan for technical assistance. We also thank Brigitta Wastuwidyaningtyas, Herman Dierick and Bryan Nikolai for helpful discussions and suggestions.

## Author contributions

D.C.M. designed, performed and discussed the experiments and wrote and edited the manuscript. C.W. designed, performed and discussed the experiments, conducted bioinformatic analysis and wrote and edited the manuscript. T.H. discussed the experiments and conducted bioinformatic analysis. T.B. discussed the experiments and edited the manuscript. B.A. helped in performing ALE experiments. R.B.N. helped in performing the P1 transductions. N.S.A. discussed the experiments and conducted bioinformatic analysis. P.K. discussed the experiments and conducted bioinformatic analysis. E.A.B helped in performing ALE experiments. T.J.C. helped in performing ALE experiments. P.D.L. helped in performing ALE experiments. S.M.R. discussed the experiments. C.H. discussed the experiments and edited the manuscript. O.L. designed and discussed the experiments and edited the manuscript.

## Competing interests

The authors declare no competing interests.
