## [Peer Review File · Nature Communications]

Reviewer comments, first round –

Reviewer #1 (Remarks to the Author):

Noteworthy Results: The authors introduce the concept of evolutionary action (EA) as a means of differentiating mutations that are having an adaptive impact as opposed to those hitchhiking with variants of adaptive significance. This method is based upon the magnitude of the amino acid substitution at the position of interest and the functional sensitivity of a sequence position to amino acid substitutions. EA is contrasted to frequency based methods of identifying variants of significance in experimental populations and clinical data. EA performed well compared to frequency methods, identifying both variants that had been previously associated with colistin and ciprofloxacin resistance, as well as some that had not been previously associated in *E. coli* K-12 MG1655.

Significance and originality: The work has tremendous significance as it provides a non-experimental method to identify important variants in clinical samples associated with antibiotic resistance (ABR), antimicrobial resistance in general, or any sort of microbial adaptation in nature. The method is original and utilizes a well conceived understanding of evolutionary theory to derive the approach.

Conclusions and claims: The conclusions and claims are well supported as the authors first confirmed that EA methods recovered variants identified by classical frequency based methods and also utilized recombinant methods to validate fitness effects found by EA predictions.

Methodology: This methodology is sound and an advancement over previously existing approaches. Details are such that the experiments can be repeated by other groups.

Reviewer #2 (Remarks to the Author):

In this manuscript named "Evolutionary Action of mutations reveals antimicrobial resistance gene", David Marciano and colleagues combine experimental evolution and multiple sequence alignment analysis (MSA) to show that antibiotic resistance mutation can be identified from the type of mutations and distant homologues. This is a very interesting approach as both in lab experiment and in epidemiological studies, inferring the causative mutations is a real challenge, especially when high mutation rate is involved. Here, the quality of the detection method supports the underlying hypothesis that antibiotic resistance mutation are harmful for the protein involved, and can then be seen as deviation from normal evolution. The comparison with frequency based approach is relevant as it is a standard way to identify mutations as convergence as the gene level has been shown to be really important (we showed that quantitatively in Tenaillon et al Science 2012).

By combining well designed experiments and epidemiological sampling and to the last author expertise on Evolutionary Trace, the authors clearly show the power of the approach and manage to recover a new gene for the resistance to colistine. Most of our comments deal with substantiating the power of this approach.

- Experimental evolution set-up: the experimental evolution experiments were led on a very large number of replicates (from 14 to 31 depending on the conditions) in 3 different conditions per antibiotic tested. This raises the question of statistical test power: is it necessary to have such a high number of replicates in order to have informative EA-KS or EA-sum gene scores? Would it be possible to obtain similar results with a smaller-scale experiment? Is the EA approach more powerful at detecting genes than the frequency-based approach when the number of experiments is low? A simple rarefaction analysis could help solve that question.

-Though the use of high mutation rate may be interesting to have more chance to get a beneficial mutation, is also come at the high cost of decreasing drastically the connection taht can be made

between mutation and the trait of interest: most mutations are passenger mutations. It is therefore possible that rather than improving the frequency based approach, the mutator decrease its power. Could it be possible to test results of both EA and frequency approach just using low mutation rate experiments. Would that explain why the *ynjC* gene was not found with the Frequency based approach? This is quite important because it is the gene that proves the benefit of EA methods. The number of mutation found in that gene if not clearly stated in the text, but a low value would give more power for the EA method.

- Throughout the article the EA approach is compared to a frequency-based approach. Would it be possible to have insights on how does it compare to an independent-site approach (such as SIFT scores) based on site conservation across distant species? Several articles have used that approach for GWASS analysis (Galardini life 2017). The question being here whether on the global scale of using MSA to do inferences, are there strong differences between the different flavours of MSA analysis to compute mutation effects.

- When performed on natural isolates, EA best-ranked genes are enriched in genes that are known to be targeted by adaptation in order to provide antibiotic resistance, however this approach is not used to detect new driver genes for antibiotic resistance. It would therefore be nice to also plot the new targets found (*ynjC*, *ouse* and even others) in experimental evolution in these graphs to see if they are sometime recruited in the wild as well. It would also be great to know what are the top ones, and to give some hints as why are they popping up in the EA approach (are they due to non random distribution of resistant strains across the phylogeny).

- A frequency-based approach in natural isolates (Figure 4) is not very adequate for comparison with EA: frequency-based approach is likely to fail due to the fact that antibiotic-resistant strains often belong to the same clade (so background mutations cannot be disentangled from adaptive mutations). This is especially true for quinolone resistance, with invading clones as ST131. The number of independent acquisitions of mutations may be a way to free oneself from this phylogenetic bias. Otherwise a SIFT score would be a good control. Bacterial GWASS as performed in Galardini could also be used, or simply connecting number of mutations in the gene to resistance and correct for phylogeny. The question is always what would be the most powerful but also the easiest to compute. Here our feeling is that the comparison to raw frequency numbers may not be fair. For instance just making the same treatment of in the EA-sum: counting frequency of mutations in resistant while not taking into accounts mutations also found in sensitive.

Minor issues:

- Introduction: "when *E. coli* grows nearly free of selective pressure, mutations are rare" => it's not so much a question of strength of selection, than a question of how adapted to its environment the strain is. If it's very well adapted, even under very strong selective pressure it won't mutate much because the number of mutations that confer a gain in fitness is low (and vice versa!).

- Results (Identification of several new driver genes for colistin resistance): "The lack of a significant effect of two tested *lpxD* mutations upon fitness in the presence of colistin may indicate (...) a role for these mutations in early stages of adaptation that becomes superseded by subsequent mutations." => This is a quite strong hypothesis. An alternative explanation may just be the lack of power due to the difficulty of measuring moderate mutation effects when some stronger effect mutation may pop in. As seen in *hemX* fitness estimates, the noise can be high if in the course of the experiment some beneficial mutations may appear in any of the backgrounds.

- Results (EA integration applied to environmental and clinical datasets): "selection for pathogenicity" => not very relevant in an *E. coli* context where disease is known to be an evolutionary dead-end, selection for increased virulence seems to be a by-product of

commensalism (genes that confer virulence also improve gut colonization).

- line 250: We do not agree with the fact that mutation improving adaptability should not be measured by the competition assay. They should, because if it fitness that is measured. Though may be such mutation should failed to be detected by EA because they are likely learning to less drastic modifications of gene functions.

Overall, it is a very promising approach with already very solid and interesting result. We just propose to substantiate a bit the conditions and reasons for its success and limits.

Olivier Tenaillon (I always sign my reviews) and an anonymous reviewer.

Reviewer #3 (Remarks to the Author):

The paper by Marciano et al. apply evolutionary action (EA) to identify driver mutations against a background of random mutations in experimental evolution experiments. They successfully identify known resistance genes to ciprofloxacin and colistin in evolved populations. Especially interesting is that EA integration was also applied to clinical and environmental resistant strains, confirming the power of the method to highly rank known drivers of resistance. The work is truly excellent and provides a much needed novel way to identify (new) resistance genes based on genome sequence and antibiotic resistance/susceptibility profiles of evolved or clinical strains. The method may be very useful for many researchers in the field. The manuscript is straightforward, reads fluently and illustrations are appropriate.

Comments:

I might have missed this information but it was unclear to me whether similar results (rankings of top hits- Referring to fig2E-H) are obtained when analyzing separately the populations having different levels of complexity (wt or wt+mutagen or mutator) instead of aggregated data? In other words, does the EA method perform equally well in complex-versus less complex populations. My main comment relates to causality of identified genes in relation to the tested resistance. I understand that mutations were restored to the wild-type sequence in evolved clones. The use of competition experiments makes perfectly sense as they correspond to the initial evolutionary selection. However, to link mutations to claimed resistance, it would be more relevant to transduce the identified mutation in a wild-type background and test for MIC, to differentiate from other fitness (and epistatic) effects. Have MIC values for ciprofloxacin-evolved (and transduced) clones been determined? Clearly, additional tests have been done on the colistin clones (some are listed in Table S2). It would make sense here also to examine at least some of the top-ranked mutations (lpxD, ybjX, ddlB, ynjC, yddW, ispB, asmA, lapB ...), also referring to Fig.5, in a wild-type background. Such an analysis is especially important when claiming that novel resistance genes have been found. This also applies to the analysis of clinical/environmental antibiotic resistant strains. Do the authors have indications that other high-ranking genes (figure 4) also affect MIC or is this rather noise that makes identification of true resistance genes difficult? Also, were novel genes contributing to for example colistin resistance identified in the latter analysis? This would clearly increase impact.

Finally, is the code publicly available and accessible to other researchers in the field?

Please provide more detail in Table S1 on the type of experiment (Cipro vs colistin) from which they were identified.

Please find below a Response to the Reviewers, detailing their comments and how each and every one of them was addressed in this revision. We are grateful for the thoughtful suggestions and appreciate the time and effort of all reviewers. We believe their input and our additions and clarifications they triggered substantially improved the manuscript and that it will now meet all expectations for acceptance and publication.

All reviewer remarks can be found below with our responses in blue text.

Reviewer #1 (Remarks to the Author):

Noteworthy Results: The authors introduce the concept of evolutionary action (EA) as a means of differentiating mutations that are having an adaptive impact as opposed to those hitchhiking with variants of adaptive significance. This method is based upon the magnitude of the amino acid substitution at the position of interest and the functional sensitivity of a sequence position to amino acid substitutions. EA is contrasted to frequency based methods of identifying variants of significance in experimental populations and clinical data. EA performed well compared to frequency methods, identifying both variants that had been previously associated with colistin and ciprofloxacin resistance, as well as some that had not been previously associated in *E. coli* K-12 MG1655.

Significance and originality: The work has tremendous significance as it provides a non-experimental method to identify important variants in clinical samples associated with antibiotic resistance (ABR), antimicrobial resistance in general, or any sort of microbial adaptation in nature. The method is original and utilizes a well conceived understanding of evolutionary theory to derive the approach.

Conclusions and claims: The conclusions and claims are well supported as the authors first confirmed that EA methods recovered variants identified by classical frequency based methods and also utilized recombinant methods to validate fitness effects found by EA predictions.

Methodology: This methodology is sound and an advancement over previously existing approaches. Details are such that the experiments can be repeated by other groups.

We thank Reviewer #1 for their encouraging response. It encapsulates the ability of EA methods to recapitulate frequency-based methods under well controlled lab conditions and improve upon identification of adaptive genes in clinical and natural isolates.

Reviewer #2 (Remarks to the Author):

In this manuscript named "Evolutionary Action of mutations reveals antimicrobial resistance gene", David Marciano and colleagues combine experimental evolution and multiple sequence alignment analysis (MSA) to show that antibiotic resistance mutation can be identified from the type of mutations and distant homologues. This is a very interesting approach as both in lab experiment and in epidemiological studies, inferring the causative mutations is a real challenge, especially when high mutation rate is involved. Here, the quality of the detection method supports the

underlying hypothesis that antibiotic resistance mutations are harmful for the protein involved, and can then be seen as deviation from normal evolution. The comparison with frequency based approach is relevant as it is a standard way to identify mutations as convergence as the gene level has been shown to be really important (we showed that quantitatively in Tenaillon et al Science 2012).

By combining well designed experiments and epidemiological sampling and to the last author expertise on Evolutionary Trace, the authors clearly show the power of the approach and manage to recover a new gene for the resistance to colistin. Most of our comments deal with substantiating the power of this approach.

We thank Reviewer 2 and his anonymous co-reviewer for their positive critique and have responded to each point below with corresponding changes to the manuscript's text.

- Experimental evolution set-up: the experimental evolution experiments were led on a very large number of replicates (from 14 to 31 depending on the conditions) in 3 different conditions per antibiotic tested. This raises the question of statistical test power: is it necessary to have such a high number of replicates in order to have informative EA-KS or EA-sum gene scores? Would it be possible to obtain similar results with a smaller-scale experiment? Is the EA approach more powerful at detecting genes than the frequency-based approach when the number of experiments is low? A simple rarefaction analysis could help solve that question.

We appreciate this suggestion from the #2 reviewers.

Yes. We performed the rarefaction/down-sampling analyses, and the results are included in a new "Evaluation of method robustness" section, Figure 4 panels E & F, and Supplementary Figures S9-S12. Our down-sampling analyses suggested that we need approximately 10 replicates in the ALE experiment to achieve similar results. There is no strong evidence indicating EA is outperforming frequency at lower sample sizes in the ALE experiment. This does however depend on the specific gene that is focused upon. Meanwhile, EA still outperforms the frequency-based method at recovering known drivers in the natural isolate dataset with just 10 isolate samples, which suggests the EA approach is generally robust.

The down-sampling analysis is described in the second paragraph of the added "Evaluation of method robustness" section which can be found at the end of the results section (page 13-14, lines 463-540):

"The experimental evolution experiments were conducted on a large number of replicates (from 14 to 31 depending on the conditions) in 3 different mutational loads per antibiotic tested. However, this large number of replicates may not be necessary to detect genes under selection. In order to address this, down-sampling analyses were performed to ascertain if similar results can be achieved with fewer samples. We first examined the number of antibiotic resistant drivers (known and newly identified) that were mutated in the subsamples (Figure S9). As expected, more drivers were mutated in the WT+mutagen and mutator conditions (8-9 for each antibiotic) compared to WT (5-6). Approximately 10-12 samples are required in order to observe at least one mutation

in each phenotypic driver. We further performed EA and frequency analyses on the subsamples and evaluated how the antibiotic resistance drivers were ranked using different sample sizes. Our methods were able to recover most drivers in the ALE datasets with less than 10 samples (Figure S10-S12). In the more complicated clinical/environmental datasets, EA integral was able to reproduce similar results as the full dataset with around 40 strains but still performed well with just 10 samples (Figure 4E&F). These findings suggested our approach is generally robust.”

-Though the use of high mutation rate may be interesting to have more chance to get a beneficial mutation, is also come at the high cost of decreasing drastically the connection taht can be made between mutation and the trait if interest: most mutations are passenger mutations. It is therefore possible that rather than improving the frequency based approach, the mutator decrease its power. Could it be possible to test results of both EA and frequency approach just using low mutation rate experiments. Would that explain why the *ynjC* gene was not found with the Frequency based approach? This is quite important because it is the gene that proves the benefit of EA methods. The number of mutation found in that gene if not clearly stated in the text, but a low value would give more power for the EA method.

Yes. We included a new Figure S4 to show the comparison of EA integration and the frequency approach for different mutational loads. There is some hint that high mutation rates decrease the power of frequency approach. EA and frequency performed similarly at low mutation rate (WT and WT+mutagen). In the mutator condition, EA is better at predicting drivers for colistin resistance but worse for ciprofloxacin resistance. If we look at the noisiest data (with the largest number of amino acid changes relative to lab strain *E. coli*), the natural isolates, EA did better at recovering known drivers for both colistin and ciprofloxacin resistance.

We added Figure S14 to show the EA distribution of *ynjC* mutations. It was only mutated once in WT+mutagen but that mutation was predicted to be impactful. This explains why EA ranked *ynjC* better in the WT+mutagen condition (Supplementary Figure 4). **The following sentence is also included in the main text Discussion section (page 16, lines 615-618).**

“The *ynjC* gene would not have been linked to colistin resistance using standard frequency-based approach, as it was ranked poorly by frequency-based approach due to the low mutation count (1 in WT+mutagen and 6 in mutator) (Figure S4 & S13). However, those mutations were predicted to be impactful by EA.”

- Throughout the article the EA approach is compared to a frequency-based approach. Would it be possible to have insights on how does it compare to an independent-site approach (such as SIFT scores) based on site conservation across distant species? Several articles have used that approach for GWASS analysis (Galardini life 2017). The question being here wether on the global scale of using MSA to do inferences, are there strong differences between the different flavours of MSA analysis to compute mutation effects.

SIFT scores and EA scores correlate well for most genes in MG1655 (see **new supplemental Figure S7**). If EA scores are substituted with SIFT scores, the top predicted genes generally remain the same (see **new supplemental Figure 8**). An exception is that using SIFT scores did not recover *ynjC* in the colistin ALE dataset. These findings suggest there are not substantial differences between the different types of MSA analyses to compute mutation effects.

We have added these results in the first paragraph of the new “Evaluation of method robustness” section found at the end of the Results section (pages 12-13, lines 438-462):

“Our results showed that the predictions of phenotype driver genes were improved over the frequency-based method, especially in the clinical/environmental datasets, when we weight the mutational impact of each amino acid substitution with its EA scores. We further examined if EA can be substituted with SIFT scores, which is commonly used to predict the impact of amino acid substitutions on protein function⁴⁹⁻⁵¹. The EA scores computed for *E. coli* K12 MG1655 show a good correlation with SIFT scores (Figure S7), with median gene level Pearson and Spearman correlations of -0.54 and -0.73, respectively. When SIFT scores are substituted for EA scores in the integration (KS test and EA-sum), the experimentally validated genes ranked similarly with the exception of *ynjC* which was not recovered in colistin ALE datasets (Figure S8). These findings suggest that the integration of mutational impact scores is a robust approach to predicting phenotype driver genes, as EA scores can be substituted by other prediction methods although this entails a slight loss of sensitivity (*ynjC*).”

- When performed on natural isolates, EA best-ranked genes are enriched in genes that are known to be targeted by adaptation in order to provide antibiotic resistance, however this approach is not used to detect new driver genes for antibiotic resistance. It would therefore be nice to also plot the new targets found (*ynjC*, *ouse* and even others) in experimental evolution in these graphs to see if they are sometime recruited in the wild as well. It would also be great to know what are the top ones, and to give some hints as why are they popping up in the EA approach (are they due to non random distribution of resistant strains across the phylogeny).

We appreciate the reviewers for this comment. We **updated Figure 4** to include newly identified genes from the ALE datasets. The newly identified drivers from the ALE were not ranked high in the natural isolate dataset. However, the top ranked genes in the natural isolate dataset clustered well in Stringdb. And there is evidence from the literature showing some of the top genes (*mukB* and *arnT*) are related to ciprofloxacin and colistin resistance.

These results are now included at the end of the “EA integration applied to environmental and clinical datasets” section (page 12, lines 428-436):

“The newly identified ciprofloxacin and colistin resistance drivers in the ALE datasets were not recovered in the clinical and environmental dataset. But, the top ranked genes by EA integral cluster significantly in Stringdb (Table S4)⁴⁵, indicating the protein products of the highly ranked genes associate with each other physically or are functionally related. For instance, *mukB* was ranked 7th by EA-KS in the ciprofloxacin environmental dataset. It was reported that *mukB* interacts with *parC*, a known

ciprofloxacin resistance driver, and stimulates the superhelical DNA relaxation activity of wild-type Topo IV⁴⁶. Another top ranked gene, *arnT* (ranked 23rd by EA-sum), in the colistin clinical dataset was reported to be activated in polymyxin/colistin resistant *E. coli*^{47,48}."

- A frequency-based approach in natural isolates (Figure 4) is not very adequate for comparison with EA: frequency-based approach is likely to fail due to the fact that antibiotic-resistant strains often belong to the same clade (so background mutations cannot be disentangled from adaptive mutations). This is especially true for quinolone resistance, with invading clones as ST131. The number of independent acquisitions of mutations may be a way to free oneself from this phylogenetic bias. Otherwise a SIFT score would be a good control. Bacterial GWASS as performed in Galardini could also be used, or simply connecting number of mutations in the gene to resistance and correct for phylogeny. The question is always what would be the most powerful but also the easiest to compute. Here our feeling is that the comparison to raw frequency numbers may not be fair. For instance just making the same treatment of in the EA-sum: counting frequency of mutations in resistant while not taking into accounts mutations also found in sensitive.

We agree with the reviewers that raw frequency number is not a good predictor for resistance gene in the natural isolate dataset. Thus, in our original analyses (both EA integration and the frequency-based approach) on the natural isolate dataset, mutations in the sensitive strains were removed from the resistance strains, so only mutations that are unique to the resistance strains were considered. Related to the above comment concerning interchangeability of SIFT scores, we also substituted EA scores with SIFT scores when performing the integration analysis on the natural isolate data (see **new Supplementary Figure S8E-H**). The known drivers for ciprofloxacin and colistin resistance were recovered at top of the gene list, suggesting accounting for mutation effects generally improves prediction in the more complicated natural isolates dataset.

Minor issues:

- Introduction: "when *E. coli* grows nearly free of selective pressure, mutations are rare" => it's not so much a question of strength of selection, than a question of how adapted to its environment the strain is. If it's very well adapted, even under very strong selective pressure it won't mutate much because the number of mutations that confer a gain in fitness is low (and vice versa!).

We agree with the reviewers that the number of mutations observed is affected by how well a strain is adapted to the environment. We were trying to express a similar idea. The "selection pressure" we meant is the relative selection pressure with respect to that specific strain. For instance, the selective pressure is high for a strain that is not adapted to ciprofloxacin if we treat it with that antibiotic. But for a strain that has *gyrA* S83L mutation, the same antibiotic treatment would manifest as a low selective pressure. **To prevent confusion, we have updated the sentence (page 2, line 63) to "when *E. coli* grows in an environment for which it is well-adapted, mutations are rare (a rate of $\sim 10^{-3}$ /genome/generation)¹⁰"**

- Results (Identification of several new driver genes for colistin resistance): "The lack of a significant effect of two tested lpxD mutations upon fitness in the presence of colistin may indicate (...) a role for these mutations in early stages of adaptation that becomes superseded by subsequent mutations." => This is a quite strong hypothesis. An alternative explanation may just be the lack of power due to the difficulty of measuring moderate mutation effects when some stronger effect mutation may pop in. As seen in hemX fitness estimates, the noise can be high if in the course of the experiment some beneficial mutations may appear in any of the backgrounds.

We appreciate this suggestion from the reviewers. We have **updated that sentence** (page 11, lines 356-359) to:

"The lack of a significant effect of two tested lpxD mutations upon fitness in the presence of colistin may indicate a limitation in the sensitivity of our competition assay especially when a stronger effect mutation in a driver gene (basS/basR) is present."

- Results (EA integration applied to environmental and clinical datasets): "selection for pathogenicity" => not very relevant in an E. coli context where disease is known to be an evolutionary dead-end, selection for increased virulence seems to be a by-product of commensalism (genes that confer virulence also improve gut colonization).

We thank the reviewers for this comment. We have **updated (page 11, line 384) "selection of pathogenicity" to "selection of commensalism"**.

- line 250: We do not agree with the fact that mutation improving adaptability should not be measured by the competition assay. They should, because if it fitness that is measured. Though may be such mutation should failed to be detected by EA because they are likely learning to less drastic modifications of gene functions.

We thank the reviewer for pointing this out. Mutations improving adaptability should be measurable by competition, but may require a longer competing time in order for new adaptive mutations to become fixed in the competing population. We have rephrased that sentence (page 11, lines 376-378):

"Also, a caveat with any of the mutations that show no fitness effects is that they may have an indirect role on resistance by increasing adaptability, which may require serial passaging over multiple days in order to be detected by the competition assay."

Overall, it is a very promising approach with already very solid and interesting result. We just propose to substantiate a bit the conditions and reasons for its success and limits.

Olivier Tenaillon (I always sign my reviews) and an anonymous reviewer.

Thank you, Dr. Tenaillon, for the thoughtful comments, they are much appreciated.

Reviewer #3 (Remarks to the Author):

The paper by Marciano et al. apply evolutionary action (EA) to identify driver mutations against a background of random mutations in experimental evolution experiments. They successfully identify known resistance genes to ciprofloxacin and colistin in evolved populations. Especially interesting is that EA integration was also applied to clinical and environmental resistant strains, confirming the power of the method to highly rank known drivers of resistance. The work is truly excellent and provides a much needed novel way to identify (new) resistance genes based on genome sequence and antibiotic resistance/susceptibility profiles of evolved or clinical strains. The method may be very useful for many researchers in the field. The manuscript is straightforward, reads fluently and illustrations are appropriate.

Thank you.

Comments:

I might have missed this information but it was unclear to me whether similar results (rankings of top hits- Referring to fig2E-H) are obtained when analyzing separately the populations having different levels of complexity (wt or wt+mutagen or mutator) instead of aggregated data? In other words, does the EA method perform equally well in complex-versus less complex populations.

Yes. The separated results for each condition are shown in Figure S2 and S4. Our methods could recover most of the drivers, but aggregating the rankings gives better results. EA and frequency performed similarly at low mutation rate (WT and WT+mutagen). In the mutator condition, EA is better at predicting drivers for colistin resistance but worse for ciprofloxacin resistance. EA did better at recovering known drivers for both colistin and ciprofloxacin resistance in the most complex clinical/environmental datasets. In order to clarify that similar results are obtained when analyzing the populations separately, we have added the text (page 6, lines 163-168):

“For the ALE of MG1655(WT), only 24 genes have nonsynonymous mutations across the 14 replicates and 3 of 5 mutated ciprofloxacin resistance drivers are the top three genes ranked by EA-sum. For the ALE of WT+mutagen, 285 genes across 15 replicates have nonsynonymous mutations and 6 of 7 mutated ciprofloxacin drivers are in the top 30 ranked genes. For the mutator strain, 3440 genes (~80% of protein coding genes) have at least one nonsynonymous mutation across 29 replicates and yet 7 of 8 known drivers are in the top 2% of ranked genes (Figure S2).”

My main comment relates to causality of identified genes in relation to the tested resistance. I understand that mutations were restored to the wild-type sequence in evolved clones. The use of competition experiments makes perfectly sense as they correspond to the initial evolutionary selection. However, to link mutations to claimed resistance, it would be more relevant to transduce the identified mutation in a wild-type background and test for MIC, to differentiate from other fitness (and epistatic) effects. Have MIC values for ciprofloxacin-evolved (and transduced) clones been determined? Clearly, additional tests have been done on the colistin clones (some are listed in Table S2). It would make sense here also to examine at least some of the top-ranked mutations (lpxD, ybjX, ddlB, ynjC, yddW, ispB, asmA, lapB ...), also referring to Fig.5, in a wild-type background. Such an analysis is especially important when claiming that novel resistance genes have been found.

We agree that re-testing the effect of mutations in the context of a wild-type background is a good experiment that will reveal whether or not the substitutions are sufficient, on their own, to confer antibiotic resistance. However, a mutation does not necessarily need to operate independently of other substitutions in order to contribute to antibiotic resistance. For instance, it has been established that initial mutations in *gyrA* provide ciprofloxacin resistance can be followed by additional mutations in *parC* that further increase ciprofloxacin resistance levels (see ref. 31 Conley et al., PLoS Pathog. 14, e1006805 [2018]). This agrees with what we observe as well in our now reported ciprofloxacin MIC data (Table S2). The *gyrA* S83L mutation is necessary to confer resistance in the evolved clone DCM292 as reversion to wild type reduces resistance to wild type levels (0.064 ug/ml) in our assay. It is also sufficient to confer resistance up to 0.512-1.024 ug/ml in the MG1655 wild type background, but it alone does not recover the full 4.096 ug/ml resistance level of the evolved DCM292 clone. In contrast, reversion of *parC* A30V in DCM292 reduces resistance levels 4-fold to 1.024 ug/ml but, in the MG1655 background, *parC* A30V does not confer any detectable resistance in our MIC assay.

We also tested several other mutations in the MG1655 wild type background including *parE* G277S, *udk* Y50C, *rstB* G195S, *yrbG* P223S, *sbcC* L173F, *mutL* A271V, *hemX* S151N, *basR* G53E, *basS* C84R, *osmE* S44N, *ynjC* P213S, *ispB* E74G, *ybjX* M73K and *lapB* E3K. With the exception of the *basS* and *basR* mutants, these substitutions do not, on their own, confer elevated resistance levels as measured by our MIC data.

We suspect that many of the mutations that contribute to fitness in the presence of antibiotics are acting epistatically with *gyrA* or *basS/R* in ciprofloxacin and colistin resistant clones, respectively. These mutants are not conferring a general growth advantage as ascertained by fitness competition results obtained in the absence of antibiotic. That data shows the revertants' fitness is either indistinguishable from the evolved clone or is actually improved in some cases (e.g. *parE* G277S, *parC* A30V, *rob* A70V and *basR* G53E). Enhanced fitness of a revertant in the absence of antibiotic suggests that under normal growth conditions there is a general fitness cost associated with those mutations (Figure S6).

We have added the results of the MIC measurements in Table S2 and added the following text (page 10, lines 303-313):

“We also examined the minimum inhibitory concentration (MIC) of ciprofloxacin for each of the revertants. Only *gyrA* S83L and *parC* A30V revertants reduced MIC levels greater than 2-fold. This is likely due to the lower sensitivity of the MIC assay relative to the competition experiments. We also transferred several of the tested mutations into the wild type MG1655 background to determine if any are sufficient, on their own, to confer ciprofloxacin resistance. Only the *gyrA* S83L mutation is sufficient to confer resistance up to 0.512-1.024 µg/ml. Also, although reversion of *parC* A30V in the evolved DCM292 strain reduces resistance levels 4-fold to 1.024 µg/ml, in the MG1655 background, *parC* A30V does not confer any detectable resistance in our MIC assay. This agrees with previous results showing *parC* mutations only increases fluoroquinolone resistance in the presence of a *gyrA* mutation (cite PMID: 8524852). Overall, the competition assay results show that we can distinguish between mutations that specifically contribute to ciprofloxacin resistance from those that have no direct fitness effects or influence fitness independently from the presence of ciprofloxacin.”

This also applies to the analysis of clinical/environmental antibiotic resistant strains. Do the authors have indications that other high-ranking genes (figure 4) also affect MIC or is this rather noise that makes identification of true resistance genes difficult?

We appreciate this comment from the reviewer. In order to address if the other top ranked genes in the natural isolate datasets are noise or not, **we performed clustering analyses with Stringdb**. The top ranked genes clustered well in Stringdb, suggesting these genes are not just random noise. There is evidence showing some of the top genes (*mukB* and *arnT*) are related to ciprofloxacin and colistin resistance.

These results are included in the “EA integration applied to environmental and clinical datasets” section with the text (page 12, lines 428-436):

“The newly identified ciprofloxacin and colistin resistance drivers in the ALE datasets were not recovered in the clinical and environmental dataset. But, the top ranked genes by EA integral cluster significantly in Stringdb (Table S4) ⁴⁵, indicating the protein products of the highly ranked genes associate with each other physically or are functionally related. For instance, *mukB* was ranked 7th by EA-KS in the ciprofloxacin environmental dataset. It was reported that *mukB* interacts with *parC*, a known ciprofloxacin resistance driver, and stimulates the superhelical DNA relaxation activity of wild-type Topo IV ⁴⁶. Another top ranked gene, *arnT* (ranked 23rd by EA-sum), in the colistin clinical dataset was reported to be activated in polymyxin/colistin resistant *E. coli* ^{47,48}.”

Also, were novel genes contributing to for example colistin resistance identified in the latter analysis? This would clearly increase impact.

We updated Figure 4 to include newly identified genes from the ALE datasets. Unfortunately, none of the newly identified phenotypic driving genes in the ALE dataset were ranked high in the clinical/environmental datasets. But the top predicted genes clustered well in Stringdb. And there is evidence from the literature showing *mukB* and *arnT* are related to ciprofloxacin and colistin resistance.

Finally, is the code publicly available and accessible to other researchers in the field?

The code is publicly available at

https://github.com/LichtargeLab/EA_antibiotics_resistance. We also have a website (<http://bioheat.lichtargelab.org>) for users to run our analysis and interactively examine the results. The EA scores for all E coli proteins are also available on that website. We have added a CODE AVAILABILITY section (page 25, lines 920-922):

“CODE AVAILABILITY

The code used in the analyses presented here can be viewed and accessed at https://github.com/LichtargeLab/EA_antibiotics_resistance.”

Please provide more detail in Table S1 on the type of experiment (Cipro vs colistin) from which they were identified.

Table S1 was updated to include this information.

Reviewer comments, second round –

Reviewer #2 (Remarks to the Author):

The authors have properly answered our comments.
Olivier Tenaillon

Reviewer #3 (Remarks to the Author):

Dear authors,

I am pleased with the responses to my concerns and have no further comments.

I would like to thank all three reviewers for their time and effort to help us improve the manuscript.

Best,

David Marciano

Reviewer #2 (Remarks to the Author):

The authors have properly answered our comments.

Olivier Tenaillon

Reviewer #3 (Remarks to the Author):

Dear authors,

I am pleased with the responses to my concerns and have no further comments.